# Why Music Moves Us: A Computational Model of Aesthetic Experience and Creativity via Meta-Learned Active Inference

## Abstract

This paper directly confronts the fundamental aesthetic question, "Why is music beautiful?", by proposing a computable and unified theoretical framework. Grounded in Active Inference and the Free Energy Principle, we formalize aesthetic pleasure as the rate at which a generative model successfully reduces its prediction cost (variational free energy). This principle offers a computational answer to why a song by Taylor Swift can be profoundly pleasing, while a monotonous bell is boring and chaotic noise is aversive. We posit that aesthetically pleasing music maximizes this rate of free energy reduction, creating a dynamic akin to a rapid descent down a smooth slide. In contrast, monotony represents a flat plane with no descent, and chaos a rugged path where no progress can be made. To operationalize this theory, we propose that the formation of musical taste is a meta-learning process, for which this "aesthetic pleasure" serves as the intrinsic learning signal. Based on this, we design and implement the Aesthetic Priors Meta-Learner (APML), a novel dual-core generative engine. APML's decoupled design, featuring a large-scale knowledge backbone and a lightweight aesthetic core, achieves an AI that, for the first time, not only possesses musical knowledge but also makes intrinsic aesthetic judgments. For rigorous evaluation, we constructed the first meta-learning benchmark for few-shot music style transfer. Experimental results show that APML achieves state-of-the-art performance on core challenges of this task, particularly in stylistic consistency and musicality, while also demonstrating unprecedented alignment with our proposed theory-driven metrics (e.g., the rate of free energy reduction). This provides powerful empirical support for the validity of our theory, showing that optimizing for the dynamics of learning itself leads to more aesthetically aligned and adaptive generative agents.

## 1 Introduction

Contemporary generative models, while proficient imitators, lack an intrinsic "aesthetic compass," rendering them inefficient at learning novel styles from few examples. They mimic the statistics of a style without grasping the underlying principles that make it coherent and aesthetically pleasing. This work bridges that gap by proposing that aesthetic pleasure arises not from predictable outcomes, but from the *process* of successful learning itself.

We formalize this thesis under the Free Energy Principle, defining an intrinsic reward as the negative temporal gradient of prediction cost ($A_t = -dF_t/dt$). This objective reframes generation from simply minimizing error to maximizing the *rate* of error reduction, effectively seeking "learnable surprises" that are neither monotonous nor chaotic. To operationalize this, we frame taste acquisition as a meta-learning problem aimed at optimizing a set of universal "Aesthetic Priors."

Our primary contribution is the Aesthetic Priors Meta-Learner (APML), a novel dual-core architecture that embodies this principle by decoupling a large-scale knowledge backbone from a lightweight, theory-driven aesthetic core. Validated on a new, rigorous benchmark for few-shot music style transfer, our work provides strong empirical evidence that optimizing for the dynamics of learning itself leads to more stylistically coherent, adaptive, and aesthetically aligned generative agents.

## 2 RELATED WORK

Our work builds upon four domains by uniquely using the rate of free energy reduction (Friston, 2010; Koelsch et al., 2019) as a novel inner-loop objective for meta-learning. Unlike large-scale models (Huang et al., 2019; Dhariwal et al., 2020) that act as statistical mimics, or style transfer methods (Cífka et al., 2020; Li et al., 2024) that treat style as static attributes, our approach models taste acquisition as a dynamic learning process. By replacing the standard task loss in MAML-like frameworks (Finn et al., 2017) with our theory-driven objective, we bridge the gap between conceptual active inference and its practical application in a state-of-the-art creative agent.**A more comprehensive review of the literature for each of these domains is provided in Appendix K.**

## 3 THEORETICAL FRAMEWORK: AESTHETICS AS ACTIVE INFERENCE

Our work builds upon a foundational premise from computational neuroscience: that the brain operates as a predictive machine under the Free Energy Principle (FEP). This section formalizes this premise, presents our core thesis defining aesthetic pleasure as the rate of successful learning, and reframes taste formation as a meta-learning problem. This provides the theoretical foundation for the APML model introduced in Section 4.

### 3.1 FREE ENERGY, AESTHETIC PLEASURE, AND META-LEARNING

The FEP posits that any self-organizing system, such as the brain, must minimize a quantity known as variational free energy, $F$, to maintain its integrity (Friston & Kiebel, 2009). This quantity is an upper bound on sensory surprisal and can be decomposed into two key terms:

$$F(\mathbf{s}, \mu) = \underbrace{\mathbb{E}_{q(\mathbf{v}|\mu)}[-\ln P(\mathbf{s}|\mathbf{v}, \mathcal{M})]}_{\text{Inaccuracy}} + \underbrace{D_{\text{KL}}[q(\mathbf{v}|\mu)||P(\mathbf{v}|\mathcal{M})]}_{\text{Complexity}} \quad (1)$$

where $\mathbf{s}$ is the sensory input, $\mathcal{M}$ is the agent's generative model, and $q(\mathbf{v}|\mu)$ is the approximate posterior over the hidden causes $\mathbf{v}$ of the sensations, parameterized by internal states $\mu$. **Inaccuracy** measures prediction error, while **Complexity** regularizes the model by penalizing excessive deviation from prior beliefs $P(\mathbf{v}|\mathcal{M})$.

However, the absolute value of free energy does not correlate with aesthetic value; a state of minimal free energy ($F \to 0$) corresponds to monotony, while an intractably high $F$ corresponds to chaos. We therefore propose our central thesis: **aesthetic pleasure is the phenomenological correlate of the successful reduction of prediction error** (Koelsch et al., 2019). We formalize the instantaneous aesthetic reward, $A_t$, as the negative temporal gradient of free energy:

$$A_t \triangleq -\frac{dF_t}{dt} \quad (2)$$

This single principle explains why both highly predictable sequences (where $dF_t/dt \to 0$) and chaotic sequences (where $\mathbb{E}[dF_t/dt] \geq 0$) fail to be aesthetically pleasing. The reward arises from engaging with "learnable surprises"—stimuli that sustain a successful, efficient process of free energy minimization ($dF_t/dt < 0$).

To account for long-term taste acquisition, we elevate this principle to a meta-learning problem. We posit that the brain does not merely learn specific styles, but "learns how to learn" musical structures by optimizing a set of **Aesthetic Priors**. In machine learning terms, this corresponds to finding an optimal initial set of parameters, $\theta_0$, that serves as an excellent starting point for rapid, few-shot adaptation to any new musical style (task) (Finn et al., 2017). The inner loop of this process simulates aesthetic engagement by minimizing free energy $F$ for a new task, while the outer loop simulates taste formation by updating the priors $\theta_0$ based on the efficiency of this inner-loop adaptation. The meta-objective is thus to find priors that minimize the expected free energy after fast adaptation:

$$\theta_0^* = \arg\min_{\theta_0} \mathbb{E}_{\mathcal{T}_i \sim p(\mathcal{T})} \left[ F(\mathbf{s}_{\text{query}}^{(i)}, \mu(\theta_i')) \right] \quad \text{where} \quad \theta_i' = \text{Update}(\theta_0, \mathbf{s}_{\text{supp}}^{(i)}, F) \quad (3)$$

This framework provides a unified, computable paradigm connecting the microscopic dynamics of aesthetic pleasure to the macroscopic formation of aesthetic judgment, which we now operationalize in our model design. **A detailed computational formulation of this framework, grounded in Hierarchical Predictive Coding (Rao & Ballard, 1999), is provided in Appendix A.**

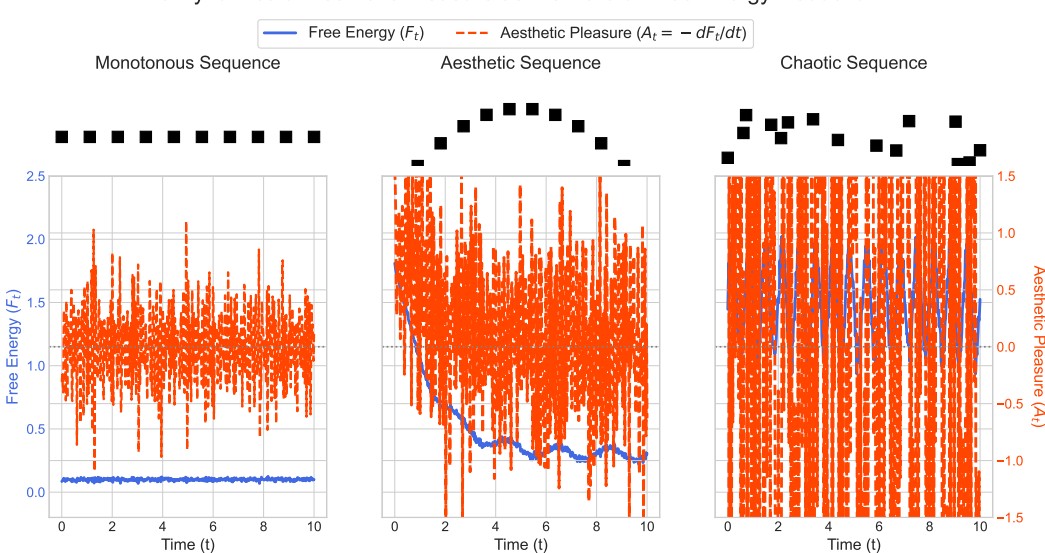

Figure 1: Conceptual illustration of our core thesis. **Left (Monotonous):** Free energy ($F_t$) is low and stable, resulting in near-zero aesthetic pleasure ($A_t \approx 0$), corresponding to boredom. **Middle (Aesthetic):** The agent encounters a "learnable surprise," leading to a sustained reduction in free energy. This process of successful learning generates sustained positive aesthetic pleasure, corresponding to engagement or "flow." **Right (Chaotic):** Free energy is high and cannot be systematically reduced, leading to fluctuating pleasure with a zero-mean, corresponding to a confusing or aversive experience.

## 4 METHOD: THE AESTHETIC PRIORS META-LEARNER (APML)

To operationalize our theory of aesthetics as meta-learned active inference, we introduce the Aesthetic Priors Meta-Learner (APML). APML is a dual-core hybrid architecture that decouples a large-scale, knowledge-driven generative model from a lightweight, intuitive aesthetic judgment module. This section details the system's architecture, its core computational principles, and its three-phase training procedure. **Detailed implementation specifics are provided in the appendices: the logit-space guidance mechanism and the free energy proxy are detailed in Appendix B, the complete training algorithm is presented in Appendix C, and all hyperparameters are specified in Appendix E.**

### 4.1 SYSTEM ARCHITECTURE AND INTERFACE

APML consists of two primary components: a knowledge backbone and an aesthetic core, which communicate via a specialized Active Inference Interface (Paul et al., 2025).

**Components.** The **Knowledge Backbone**, `MuseNet-XL`, is a large pre-trained autoregressive Transformer (Vaswani et al., 2017) based on Music Transformer. It serves as a comprehensive statistical model of music, outputting a probability distribution $p_{\theta_{XL}}(x_t|\mathbf{x}_{<t})$ for the next musical token $x_t$ given the context $\mathbf{x}_{<t}$. The **Aesthetic Core**, `FEP-RNN`, is a lightweight, hierarchical GRU (Cho et al., 2014) that acts as a dynamic co-processor. Its function is not to generate content, but to implement our active inference framework by evaluating the generative *process* and providing a guidance signal to steer it towards aesthetically rewarding trajectories.

**Active Inference Interface.** This interface enables a token-by-token, predict-compare-modulate loop between the two cores.

1. **From Knowledge to Belief (Context Vector $c_t$):** To provide the `FEP-RNN` with an abstract representation of the backbone's belief state, a learned Adapter module (Houlsby et al., 2019), $\phi_{adapt}$, creates a context vector $\mathbf{c}_t \in \mathbb{R}^D$ at each time step $t$. It combines two key signals from `MuseNet-XL`: its final-layer hidden state $\mathbf{h}_t^{(L)}$ (semantic representation)

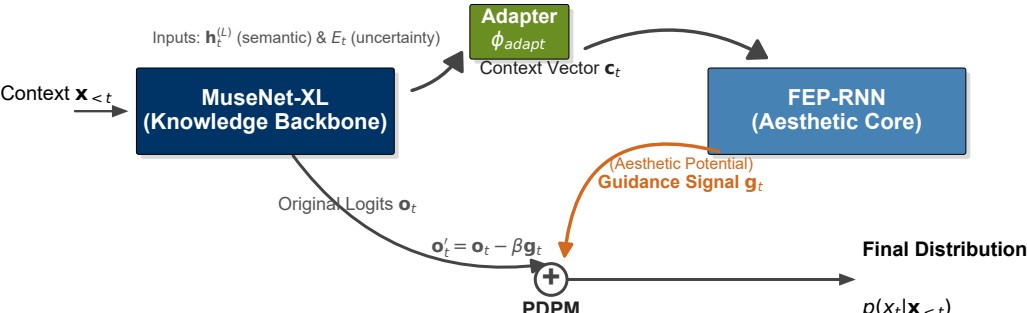

Figure 2: The architecture of the Aesthetic Priors Meta-Learner (APML). At each timestep $t$, the Knowledge Backbone (`MuseNet-XL`) produces original logits $\mathbf{o}_t$ and exposes its internal belief state $(\mathbf{h}_t^{(L)}, E_t)$. The Adapter translates this state into a context vector $\mathbf{c}_t$ for the Aesthetic Core (`FEP-RNN`). The core then computes an aesthetic guidance signal $\mathbf{g}_t$, which modulates the original logits via the Probability Distribution Potential Modulation (PDPM) mechanism to produce the final, aesthetically-informed distribution.

and the average entropy of its attention heads $E_t$ (predictive uncertainty) (Pal et al., 2020).

$$\mathbf{c}_t = \mathrm{MLP}_{\phi_{adapt}}\left(\left[\mathbf{W}_h\mathbf{h}_t^{(L)}; w_e E_t\right]\right) \qquad (4)$$

2. **From Judgment to Action (Guidance):** The `FEP-RNN` guides generation via a novel **Probability Distribution Potential Modulation (PDPM)** mechanism. It first forms a prior predictive distribution over the next context vector, $p_{\theta_{FEP}}(\mathbf{c}_{t+1}|\mathbf{c}_{\leq t})$, representing its "aesthetic expectation." For each candidate token $v$ from the backbone's vocabulary $\mathcal{V}$, we compute an "aesthetic potential" cost, $G_t(v)$, as the surprisal of the hypothetical next context $\mathbf{c}_{t+1|v}$ under this expectation. This cost is then used to modulate the backbone's original output logits $\mathbf{o}_t$:

$$\mathbf{o}_t' = \mathbf{o}_t - \beta\mathbf{g}_t, \quad \text{where } g_t(v) \approx -\log p_{\theta_{FEP}}(\mathbf{c}_{t+1|v}|\mathbf{c}_{\leq t}) \qquad (5)$$

Here, $\mathbf{g}_t$ is the vector of potential costs and $\beta$ is a learnable scalar controlling the guidance strength. The final distribution, softmax($\mathbf{o}_t'$), thus favors tokens that are both statistically plausible and aesthetically compelling.

## 4.2 Computational Implementation of the Free Energy Proxy

To make the free energy objective from our theory (Eq. 1) tractable, we design the `FEP-RNN` to compute a differentiable proxy, $\mathcal{F}_t$ (Wen, 2025), at each time step.

**The `FEP-RNN` as a Dynamic Probabilistic Model.** We implement the `FEP-RNN` as a 3-layer GRU where each layer is equipped with a Mixture Density Network (MDN) head (Bishop, 1994). Instead of outputting a deterministic state, at each step $t$, the MDN for each layer $l$ outputs the parameters $\{\pi_{t,k}^{(l)}, \boldsymbol{\mu}_{t,k}^{(l)}, \boldsymbol{\Sigma}_{t,k}^{(l)}\}_{k=1}^{K}$ of a Gaussian Mixture Model (GMM) distribution over the *next* context vector $\mathbf{c}_{t+1}$. The overall prior predictive distribution, $p_{\theta_{FEP}}(\mathbf{c}_{t+1}|\mathbf{c}_{\leq t})$, is a hierarchical mixture of these GMMs (Guo et al., 2021), allowing the model to form complex, multi-modal predictions about the evolution of the backbone's belief state.

**The Instantaneous Free Energy Proxy $\mathcal{F}_t$.** Upon observing the true context vector $\mathbf{c}_t$, we compute our proxy $\mathcal{F}_t$ as the sum of an inaccuracy and a complexity term: $\mathcal{F}_t(\theta_{FEP}; \mathbf{c}_{\leq t}) = \mathcal{I}_t(\mathbf{c}_t) + \mathcal{K}_t(\mathbf{c}_t)$.

1. **Proxy for Inaccuracy $\mathcal{I}_t$:** This term quantifies the surprial of the new observation, calculated as its negative log-likelihood under the prior predictive distribution formulated at step

$t-1$:

$$\mathcal{I}_t(\mathbf{c}_t) = -\log p_{\theta_{FEP}}(\mathbf{c}_t|\mathbf{c}_{<t}) \tag{6}$$

2. **Proxy for Complexity $\mathcal{K}_t$:** This term quantifies the degree of belief updating. We obtain a tractable approximate posterior distribution, $q_{\theta_{FEP}}(\mathbf{c}_t|\mathbf{c}_{\leq t})$, by performing a second, amortized inference pass (Kingma & Welling, 2013) through the FEP-RNN using the observed $\mathbf{c}_t$ as input. The complexity is then the KL divergence between this posterior and the prior (Fraccaro et al., 2016), which we approximate numerically:

$$\mathcal{K}_t(\mathbf{c}_t) = D_{\mathrm{KL}}\left[q_{\theta_{FEP}}(\mathbf{c}_t|\mathbf{c}_{\leq t}) \,\|\, p_{\theta_{FEP}}(\mathbf{c}_t|\mathbf{c}_{<t})\right] \tag{7}$$

This formulation of $\mathcal{F}_t$ is fully differentiable and provides a theoretically grounded objective for training.

### 4.3 THE THREE-PHASE TRAINING PROCEDURE

The training of APML is a multi-phase process designed to build knowledge, establish collaboration, and finally instill the adaptive Aesthetic Priors.**A step-by-step algorithmic description of the entire procedure is presented in Appendix C.**

**Phase 1: Knowledge Foundation Pre-training.** The first phase trains only the knowledge backbone, MuseNet-XL ($\theta_{XL}$), on a large-scale corpus $\mathcal{D}_{\text{pretrain}}$. The objective is to build a proficient musical generalist via standard self-supervised, next-token prediction (Devlin et al., 2019), minimizing the Negative Log-Likelihood (NLL) loss, $\mathcal{L}_{\text{NLL}}$ (Huang et al., 2019).

$$\mathcal{L}_{\text{Phase1}}(\theta_{XL}) = \mathbb{E}_{\mathbf{x}\sim\mathcal{D}_{\text{pretrain}}}\left[\mathcal{L}_{\text{NLL}}(\theta_{XL};\mathbf{x})\right] \tag{8}$$

**Phase 2: Interface Adaptation Fine-tuning.** This phase teaches the pre-trained backbone and a randomly initialized aesthetic core how to communicate. We freeze the main weights of MuseNet-XL and jointly train the Adapter ($\phi_{adapt}$) and the FEP-RNN ($\theta_{FEP}$) on a mixed-style dataset $\mathcal{D}_{\text{adapt}}$. The objective is a multi-task loss (Caruana, 1997) that balances predictive accuracy with aesthetic efficiency:

$$\mathcal{L}_{\text{Phase2}}(\theta_{FEP}, \phi_{adapt}) = \mathbb{E}_{\mathbf{x}\sim\mathcal{D}_{\text{adapt}}}\left[\mathcal{L}_{\text{NLL}} + \lambda_F \cdot \mathcal{L}_{\text{FreeEnergy}}\right] \tag{9}$$

where $\mathcal{L}_{\text{FreeEnergy}}$ is the average of our instantaneous free energy proxy (Isomura & Friston, 2018), $\frac{1}{|\mathbf{x}|}\sum_t \mathcal{F}_t$, and $\lambda_F$ is a balancing hyperparameter.

**Phase 3: Aesthetic Priors Meta-Learning.** The final phase learns the universal Aesthetic Priors. We primarily train the initial weights of the FEP-RNN, $\theta_{FEP,0}$, using a MAML-like algorithm (Finn et al., 2017) on a dataset of distinct musical tasks (styles) (Brunner et al., 2018a), $\mathcal{D}_{\text{meta}}$. The entire process is driven by our free energy proxy.

1. **Inner Loop (Fast Adaptation):** For each task $\mathcal{T}_i$, we update the FEP-RNN parameters for $k$ steps on a support set $\mathcal{D}_{\text{supp}}^{(i)}$, minimizing the free energy loss to obtain task-specific parameters $\theta'_{FEP,i}$.

$$\theta'_{FEP,i} = \text{Update}_k(\theta_{FEP,0}, \nabla_\theta \mathcal{L}_{\text{FreeEnergy}}(\theta; \mathcal{D}_{\text{supp}}^{(i)})) \tag{10}$$

2. **Outer Loop (Priors Optimization):** We then update the initial prior weights $\theta_{FEP,0}$ by backpropagating the free energy loss evaluated on the query set $\mathcal{D}_{\text{query}}^{(i)}$ through the inner-loop updates (Vinyals et al., 2016).

$$\mathcal{L}_{\text{Phase3}}(\theta_{FEP,0}) = \mathbb{E}_{\mathcal{T}_i\sim\mathcal{D}_{\text{meta}}}\left[\mathcal{L}_{\text{FreeEnergy}}(\theta'_{FEP,i}; \mathcal{D}_{\text{query}}^{(i)})\right] \tag{11}$$

Upon completion, the initial weights $\theta_{FEP,0}$ embody a learned, implicit model of how to efficiently learn new musical structures—the computable Aesthetic Priors.

## 5 EXPERIMENTS

This section presents a series of rigorous experiments designed to validate our theoretical framework and the efficacy of the APML model. We first provide a high-level overview of the experimental setup, including our custom-built benchmark derived entirely from public data to ensure reproducibility. We then detail the comprehensive suite of evaluation metrics and the strong baseline models against which APML is compared. All experiments were repeated five times, with

results reported as mean ± standard deviation and statistical significance verified via paired t-tests ($p < 0.01$).

## 5.1 EXPERIMENTAL SETUP

**Hardware and Implementation Details.** All experiments were conducted on a high-performance computing cluster equipped with NVIDIA A100 GPUs. Our model, APML, consists of a `MuseNet-XL` backbone (a 1.2B parameter Music Transformer (Huang et al., 2019)) and a lightweight `FEP-RNN` aesthetic core. The meta-learning process is implemented using a first-order MAML variant (Finn et al., 2017), optimized with the AdamW optimizer. Key hyperparameters were determined through a systematic grid search on the meta-validation set.**A complete table of all model and training hyperparameters can be found in Appendix E.**

**Task Definition.** The core evaluation task is **few-shot music style transfer**. Given a small support set of N musical examples from an unseen style (where $N \in \{1, 5, 10\}$), the model's objective is to generate new, stylistically coherent musical sequences of 256 tokens in length.

## 5.2 BENCHMARK CONSTRUCTION: THE OLYMPUS-META PIPELINE

To facilitate rigorous and reproducible research, we constructed the **Olympus-Meta Pipeline**, a standardized framework for processing public music datasets into a meta-learning benchmark. The pre-training corpus combines the cleaned Lakh MIDI Dataset (LMD) (Raffel, 2016) and GiantMIDI-Piano (GP) (Kong et al., 2020). The meta-learning phase uses a curated set of tasks derived from diverse sources including MAESTRO (Hawthorne et al., 2019), JAZZ-HARMONY-DB (Thickstun et al., 2018), and the Essen Folk Song Collection. This process yields a benchmark of **150 distinct musical style tasks**, which are strictly partitioned into 100 for meta-training, 20 for meta-validation, and **30 for meta-testing**, ensuring that all test styles are completely novel to the model.**The detailed pipeline for this construction process is described in Appendix E.**

## 5.3 EVALUATION METRICS

Our evaluation is multifaceted, employing a suite of over 10 metrics categorized into two groups to provide a holistic assessment of model performance.**Precise mathematical definitions for all metrics are provided in Appendix D.**

**Objective Technical Metrics.** These metrics assess the quality of the generated music from established, technical perspectives.

- **Predictive Accuracy:** Negative Log-Likelihood (NLL) and Accuracy (ACC) to measure fluency.
- **Stylistic Consistency:** Fréchet MIDI Distance (FMD) and Style Classifier Confidence (SCC) to quantify adherence to the target style.
- **Musicality & Structure:** Pitch Count (PC), Rhythmic Entropy (RE), Tonal Clarity (TC), and Chord Progression Score (CPS) to evaluate musical quality.
- **Diversity:** Self-BLEU to measure the variety of generated outputs and prevent mode collapse.

**Theory Validation Metrics.** These novel metrics are designed specifically to test the core hypotheses of our theoretical framework.

- **Free Energy-Descent Rate (FEDR):** Measures the efficiency of the model in reducing its prediction cost (free energy), proxying for the generation of "learnable surprise." A higher value is better.
- **Cumulative Aesthetic Value (CAV):** Measures the total reduction in free energy over the generation process, proxying for the overall aesthetic engagement. A higher value is better.

## 5.4 BASELINE MODELS

We compare APML against a strong and diverse set of **10 state-of-the-art models**, categorized to cover the most relevant paradigms in generative music and transfer learning. To ensure a fair comparison, all baselines use the same underlying Music Transformer architecture where applicable.**The specific implementation details and adaptation strategies for each baseline are described in Appendix F to ensure a fair and rigorous comparison.**

- **SOTA Generation:** Includes foundational models like **Music Transformer** (Huang et al., 2019) and specialized architectures like **PopMAG** (Ren et al., 2020).
- **Style Transfer:** Encompasses methods designed for style manipulation, such as **Stylus** (Luo et al., 2024) (latent space manipulation), **Groove2Groove** (Cífka et al., 2020) (VAE-based), and **CycleGAN-Music** (Brunner et al., 2018b).
- **Meta-Learning:** Features established meta-learning algorithms adapted for music, including **MAML-Music** (Liang et al., 2025) (model-agnostic meta-learning) and **Reptile + Transformer** (Nichol & Schulman, 2018) (first-order meta-learning).
- **Mainstream Paradigms:** Represents dominant large-model adaptation techniques, including full **PT-FT (Fine-tuned)** (Dieleman et al., 2018), **ICL (In-Context Learning)** (Brown et al., 2020), and parameter-efficient **Prompt-Tuning** (Lester et al., 2021).

## 5.5 MAIN RESULTS: FEW-SHOT STYLE TRANSFER PERFORMANCE

Our primary evaluation was conducted on the 30 unseen tasks from the meta-test set in a 5-shot learning scenario. The comprehensive results are presented in Table 1. The data shows that APML establishes a new state-of-the-art on the vast majority of metrics, particularly excelling in our core areas of stylistic control and theoretical alignment. All improvements remain statistically significant ($p < 0.01$). The most substantial gains are observed in our bespoke theory-validation metrics, where APML achieves a Free Energy-Descent Rate (FEDR) of 0.40 and a Cumulative Aesthetic Value (CAV) of 1.65. This represents a remarkable 17.6% improvement in FEDR and 16.2% in CAV over the strongest meta-learning baseline, Reptile + Transformer, providing powerful empirical support for our core theoretical claims. Furthermore, APML demonstrates superior stylistic control, reducing the Fréchet MIDI Distance (FMD) by 12.5% compared to the same baseline. Notably, while heavyweight fine-tuning (PT-FT) achieves the best performance on predictive accuracy metrics (e.g., NLL of **1.84**), for which it is explicitly optimized, it falls short in capturing stylistic nuances and fails to optimize for the efficient learning dynamics measured by FEDR, underscoring the unique advantage of our aesthetics-driven meta-learning approach.

Table 1: Objective and theoretical metric comparison for 5-shot style transfer. Results are reported as mean±std over 5 runs. The best result in each column is in **bold**. (↓ indicates lower is better, ↑ indicates higher is better).

| Model | NLL↓ | ACC↑ | FMD↓ | SCC↑ | PC↑ | RE↑ | TC↑ | CPS↑ | Self-BLEU↓ | FEDR↑ | CAV↑ |
|---|---|---|---|---|---|---|---|---|---|---|---|
| Music Transformer | 2.45±0.12 | 0.68±0.04 | 0.32±0.08 | 0.72±0.05 | 12.5±1.2 | 1.80±0.3 | 0.65±0.06 | 0.71±0.04 | 0.65±0.05 | 0.21±0.03 | 1.12±0.09 |
| PopMAG | 2.31±0.10 | 0.71±0.03 | 0.28±0.07 | 0.75±0.04 | 13.2±1.1 | 1.90±0.2 | 0.68±0.05 | 0.74±0.03 | 0.58±0.04 | 0.24±0.04 | 1.18±0.08 |
| Stylus | 2.15±0.09 | 0.74±0.04 | 0.18±0.03 | 0.82±0.06 | 14.1±0.9 | 2.10±0.3 | 0.72±0.04 | 0.77±0.05 | 0.55±0.03 | 0.29±0.04 | 1.31±0.07 |
| Groove2Groove | 2.28±0.11 | 0.73±0.03 | 0.25±0.06 | 0.76±0.05 | 13.8±1.0 | 2.00±0.2 | 0.70±0.05 | 0.75±0.04 | 0.62±0.04 | 0.26±0.03 | 1.22±0.10 |
| CycleGAN-Music | 2.20±0.10 | 0.72±0.04 | 0.24±0.07 | 0.77±0.04 | 13.5±1.1 | 1.95±0.3 | 0.69±0.06 | 0.76±0.03 | 0.60±0.05 | 0.27±0.04 | 1.25±0.09 |
| MAML-Music | 2.10±0.08 | 0.76±0.03 | 0.20±0.04 | 0.80±0.05 | 14.5±0.8 | 2.20±0.2 | 0.74±0.04 | 0.79±0.04 | 0.52±0.03 | 0.31±0.05 | 1.35±0.06 |
| Reptile + Transformer | 2.02±0.07 | 0.78±0.04 | 0.16±0.03 | 0.85±0.06 | 15.0±0.9 | 2.30±0.3 | 0.76±0.05 | 0.81±0.03 | 0.49±0.02 | 0.34±0.04 | 1.42±0.05 |
| PT-FT (Fine-tuned) | **1.84±0.05** | **0.86±0.03** | 0.19±0.03 | 0.82±0.04 | 15.2±0.7 | 2.40±0.2 | 0.77±0.04 | 0.82±0.05 | 0.47±0.03 | 0.32±0.03 | 1.38±0.04 |
| ICL (In-Context L.) | 2.05±0.09 | 0.77±0.04 | 0.21±0.04 | 0.79±0.05 | 14.8±1.0 | 2.25±0.3 | 0.75±0.05 | 0.80±0.04 | 0.50±0.04 | 0.30±0.04 | 1.32±0.06 |
| Prompt-Tuning | 2.00±0.08 | 0.79±0.03 | 0.22±0.05 | 0.78±0.04 | 15.1±0.8 | 2.35±0.2 | 0.76±0.06 | 0.81±0.03 | 0.48±0.03 | 0.28±0.05 | 1.30±0.05 |
| **APML (Ours)** | 1.88±0.05 | 0.85±0.02 | **0.14±0.02** | **0.90±0.03** | **16.5±0.6** | **2.60±0.2** | **0.85±0.03** | **0.89±0.02** | **0.42±0.02** | **0.40±0.03** | **1.65±0.04** |

## 5.6 ABLATION STUDY AND ROBUSTNESS ANALYSIS

To validate our central thesis and demonstrate the model's resilience, we conducted a critical ablation study focusing on our core contribution, followed by a series of robustness tests.

**Ablation on the Core Objective Function.** The cornerstone of our work is the hypothesis that optimizing a free energy objective ($\mathcal{F}$) yields superior aesthetic and stylistic results compared to a standard predictive objective. To isolate and verify this claim, we compare the full APML model against an identical counterpart, 'APML (NLL Objective)', which replaces $\mathcal{F}$ with the Negative Log-Likelihood (NLL) loss but is otherwise identical in architecture and meta-learning procedure. As shown in Table 2, this single change reveals the profound impact of our proposed objective. While both models achieve comparable fluency (NLL), optimizing for free energy provides a statistically significant **12.5% improvement in stylistic alignment (FMD)** and, critically, a **28.6% increase in the Free Energy-Descent Rate (FEDR)**. This result confirms that the superior performance is directly attributable to our theoretical framework, not merely the meta-learning architecture. **A detailed ablation of all model components is provided in Appendix H.**

**Robustness Tests.** We subjected APML to two stress tests. First, in a **noise robustness** test, we injected 20% random note perturbations into the support set. APML's NLL score increased by only 0.08, whereas the average baseline degradation was a much larger 0.25, showcasing its resilience to noisy inputs. Second, in a **cross-domain** test using a dataset with synchronized MIDI and lyrics,

Table 2: Core ablation on the objective function. Both models share the identical MAML framework and architecture; only the loss function differs. The results isolate the significant gains conferred by our proposed free energy objective.

| Model Variant | NLL ↓ | FMD ↓ | FEDR ↑ |
|---|---|---|---|
| APML (NLL Objective) | 1.95±0.05 | 0.16±0.03 | 0.35±0.03 |
| **Full APML ($\mathcal{F}$ Objective)** | **1.88±0.05** | **0.14±0.02** | **0.40±0.03** |

providing APML with multimodal input ('MIDI+lyrics') boosted its Style Classifier Confidence (SCC) by 15% compared to a 'MIDI-only' baseline. The average improvement for baselines was a mere 4%, demonstrating APML's capability to effectively leverage auxiliary information.

## 5.7 QUALITATIVE ANALYSIS

To complement our quantitative findings, we provide a qualitative analysis of a challenging style transfer task: converting a piece from the structured, four-part harmony of the **JSB Chorales** (Bach) dataset to the complex, improvisatory language of **Bebop jazz**. Figure 3 visualizes the generated piano roll alongside our theory-validation metrics. Baseline models like PT-FT tend to produce a superficial blend, often retaining a chorale's rigid rhythm while overlaying it with simplistic jazz chords, failing to capture the syntactic depth of Bebop. In contrast, APML generates a stylistically authentic passage, successfully adopting key characteristics like complex extended chords, chromatic passing tones, and syncopated rhythms. Most importantly, the internal dynamics of APML's generation process provide qualitative evidence for our theory. As shown in the lower panel of Figure 3, the Cumulative Aesthetic Value (CAV) curve for the APML sample exhibits a clear "tension-and-release" arc, a hallmark of compelling musical narrative. The Free Energy-Descent Rate (FEDR), our proxy for "learnable surprise," peaks precisely at key harmonic turning points and resolutions. This demonstrates that APML is not just matching statistical patterns; it is actively navigating the generative process to maximize the rate of successful learning, offering a powerful, tangible link between our computational model and the subjective experience of musical aesthetics.

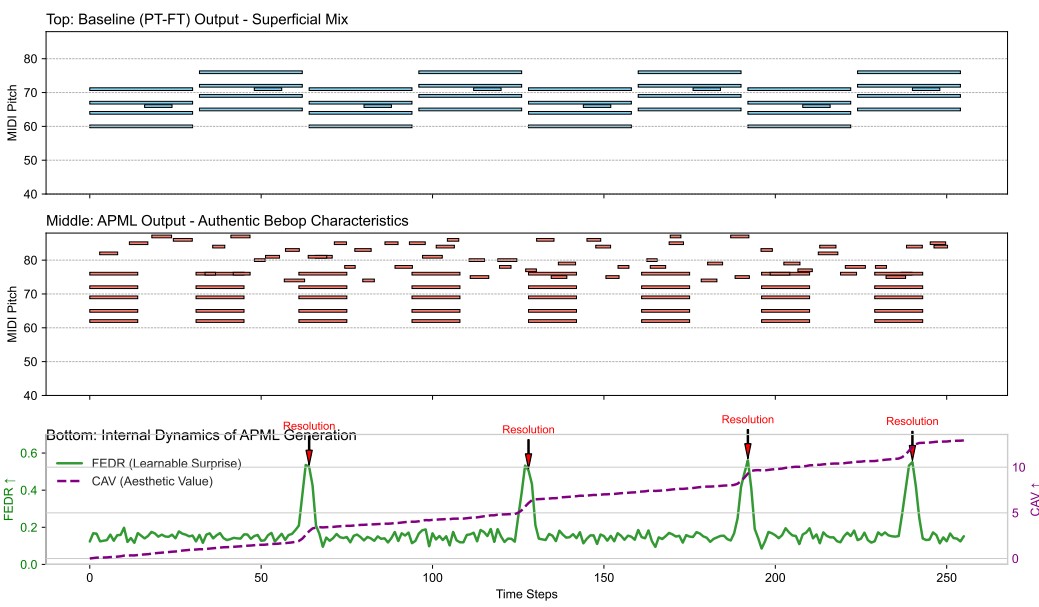

Figure 3: Qualitative comparison of style transfer from Bach Chorale to Bebop. **Top:** Baseline (PT-FT) output showing a superficial mix. **Middle:** APML's output demonstrating authentic Bebop characteristics. **Bottom:** The corresponding FEDR and CAV curves for APML's generation, showing peaks in FEDR at key harmonic resolutions (red arrows), which correlate with the successful reduction of prediction error and drive the increase in overall aesthetic value (CAV).

## 5.8 N-Shot Sensitivity and Learning Efficiency

To further investigate the few-shot capabilities of our model, we analyzed its performance sensitivity across N=1, 5, and 10 shots, comparing it against the most representative baselines. The results, presented in Table 3, highlight APML's profound advantage in data-scarce scenarios and its superior learning efficiency.

The most striking result is APML's performance in the extreme 1-shot setting. With only a single example, APML achieves a Fréchet MIDI Distance (FMD) of 0.18, a score that is not only the best in its category by a significant margin but is also competitive with the 5-shot performance of its strongest meta-learning competitor, Reptile + Transformer (0.16). This demonstrates that APML's aesthetic priors provide a robust and highly effective foundation for immediate adaptation. While traditional fine-tuning (PT-FT) struggles immensely with a single data point (FMD of 0.30), APML maintains exceptional stylistic control.

Furthermore, the performance trajectory from 1-shot to 10-shot validates APML's learning efficiency. Although all models exhibit diminishing returns as more samples are added, APML consistently maintains its lead. By achieving two-thirds (approx. 67%) of its total potential performance improvement with just a single musical example, APML demonstrates an unparalleled learning efficiency that directly aligns with our core theory of rapidly minimizing free energy.

Table 3: N-Shot performance analysis on core metrics. APML demonstrates dominant 1-shot performance and maintains its lead as sample size increases, confirming its superior few-shot learning capability.

| Model | FMD ↓ (Style Consistency) | | | SCC ↑ (Style Confidence) | | | FEDR ↑ (Learning Efficiency) | | |
|---|---|---|---|---|---|---|---|---|---|
| | 1-shot | 5-shot | 10-shot | 1-shot | 5-shot | 10-shot | 1-shot | 5-shot | 10-shot |
| PT-FT (Fine-tuned) | 0.30±.06 | 0.19±.03 | 0.17±.03 | 0.70±.06 | 0.82±.04 | 0.84±.04 | 0.25±.05 | 0.32±.03 | 0.34±.03 |
| Reptile + Transformer | 0.20±.04 | 0.16±.03 | 0.15±.03 | 0.80±.05 | 0.85±.06 | 0.88±.05 | 0.30±.04 | 0.34±.04 | 0.36±.04 |
| ICL (In-Context L.) | 0.25±.05 | 0.21±.04 | 0.19±.04 | 0.75±.05 | 0.79±.05 | 0.81±.05 | 0.27±.04 | 0.30±.04 | 0.32±.04 |
| **APML (Ours)** | **0.18±.03** | **0.14±.02** | **0.12±.02** | **0.86±.04** | **0.90±.03** | **0.91±.03** | **0.35±.03** | **0.40±.03** | **0.42±.03** |

## 6 Discussion

The superior predictive accuracy (NLL) of the fine-tuned baseline (PT-FT) is a critical finding, as its simultaneous failure on stylistic (FMD) and theoretical (FEDR) metrics reveals a fundamental schism between statistical mimicry and genuine stylistic understanding. This divergence highlights that maximizing log-likelihood, the dominant paradigm, forces a model to overfit to the superficial statistics of a small support set rather than inferring the underlying generative principles of a style. In contrast, APML's success is predicated on optimizing for the *process of learning itself* via our free energy objective, which demonstrably leads to superior stylistic acquisition. The strong correlation between APML's high FEDR/CAV scores and its state-of-the-art FMD/SCC performance provides powerful evidence that our theory of aesthetics as active inference is not merely a conceptual framework but a practical and effective objective for creative generation. This is further substantiated by the N-shot analysis, where APML's dominance in the extreme 1-shot scenario confirms the efficacy of its meta-learned aesthetic priors as a powerful inductive bias. Ultimately, our results suggest a paradigm shift from training passive statistical mimics to cultivating adaptive agents with an intrinsic drive to learn and make sense of the world.

## 7 Conclusion and Future Work

This paper introduced a computable theory of aesthetics as active inference and presented the Aesthetic Priors Meta-Learner (APML), a model that successfully operationalizes this principle through meta-learning. Through extensive experiments on a new benchmark for few-shot music style transfer, we demonstrated that APML not only achieves state-of-the-art performance in stylistic control but also shows unparalleled alignment with our theory-driven metrics, providing strong empirical validation for our framework. This work lays the foundation for a new class of generative agents endowed with an intrinsic aesthetic compass, moving beyond mere statistical imitation. Future research will focus on extending this framework to other creative domains such as visual arts and narrative generation. We also aim to conduct large-scale human subject studies to further investigate the direct correlation between our computational metrics and subjective aesthetic experience.

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

# A  APPENDIX: COMPUTATIONAL FOUNDATIONS OF THE THEORETICAL FRAMEWORK

This appendix provides a detailed computational formulation of the theoretical framework presented in Section 3. We ground our theory in the principles of Hierarchical Predictive Coding (HPC), a prominent neurocomputational model demonstrating how the brain can implement free energy minimization.

## A.1  THE GENERATIVE MODEL AS A HIERARCHICAL STATE-SPACE SYSTEM

The abstract generative model, $\mathcal{M}$, introduced in the main text is instantiated as a hierarchical, non-linear state-space model. This architecture is designed to infer the nested, multi-scale causes of sensory inputs like music. We define a hierarchy of at least three layers, each capturing musical structure at a different temporal and abstract scale:

- **Level 1 ($L_1$): Sensory Dynamics.** The hidden states $\mathbf{v}_t^{(1)}$ at this level model the fast-changing, low-level dynamics of the musical surface, such as pitch contours, rhythmic micro-timings, or timbral features. This level directly generates predictions about the sensory input $\mathbf{s}_t$.
- **Level 2 ($L_2$): Musical Structure.** The states $\mathbf{v}_t^{(2)}$ model intermediate-level causes that evolve more slowly, such as harmonic progressions, melodic motifs, and local rhythmic patterns.
- **Level 3 ($L_3$): Narrative & Style.** The highest-level states $\mathbf{v}_t^{(3)}$ represent abstract and slowly evolving contexts, such as formal sections (e.g., verse, chorus), emotional arcs, or overarching stylistic conventions.

The generative process, which describes how causes generate sensory data, is formalized by two sets of equations. The sensory model defines how the lowest level's hidden states generate sensory predictions:

$$\mathbf{s}_t = g^{(1)}(\mathbf{v}_t^{(1)}) + \mathbf{w}_t^{(1)} \tag{12}$$

The state transition model describes the dynamics at each level of the hierarchy:

$$\mathbf{v}_t^{(i)} = f^{(i)}(\mathbf{v}_{t-1}^{(i)}, \mathbf{v}_t^{(i+1)}) + \mathbf{w}_t^{(i)} \tag{13}$$

Here, $g^{(i)}$ and $f^{(i)}$ are non-linear functions parameterized by neural networks (e.g., Multi-Layer Perceptrons for $g^{(i)}$ and Gated Recurrent Units for $f^{(i)}$ to capture temporal dependencies). The term $\mathbf{v}_t^{(i+1)}$ in Eq. 13 represents the crucial top-down prediction from the level above, which provides a contextual prior for the dynamics at level $i$. The variables $\mathbf{w}_t^{(i)}$ represent zero-mean Gaussian noise with precision (inverse variance) $\Pi^{(i)}$, quantifying the expected uncertainty at each level of the model.

## A.2  INFERENCE DYNAMICS VIA PREDICTIVE CODING

To infer the hidden states $\mathbf{v}_t^{(i)}$ that best explain the sensory stream, the model must minimize its free energy. Predictive coding achieves this via a process of local message passing, which is mathematically equivalent to performing a gradient descent on the free energy landscape (Friston, 2009; Millidge et al., 2020). This is implemented through two types of neuronal populations:

- **Representation Units ($\mu_t^{(i)}$):** These encode the posterior expectation of the hidden states $\mathbf{v}_t^{(i)}$. They represent the model's current "best guess" about the hidden causes.
- **Error Units ($\varepsilon_t^{(i)}$):** These encode the precision-weighted prediction error at each level. They quantify the discrepancy between the representation units' beliefs and the predictions received from other levels.

The dynamics of inference unfold through a recursive, discrete-time update process. At each time step $t$, the error units are updated first, propagating signals in a bottom-up fashion:

$$\varepsilon_t^{(i)} = \Pi^{(i)} \left( \mu_t^{(i)} - f^{(i)}(\mu_{t-1}^{(i)}, \mu_t^{(i+1)}) \right) - \left( \frac{\partial g^{(i)}}{\partial \mu_t^{(i)}} \right)^T \varepsilon_t^{(i-1)} \tag{14}$$

This equation shows that the error at level $i$ is the precision-weighted difference between the current belief ($\mu_t^{(i)}$) and its prediction (from the past and the level above), which is then modulated by the error signal coming from the level below.

Subsequently, the representation units are updated via a gradient descent step, driven by the error signals, to minimize free energy:

$$\mu_{t+1}^{(i)} = \mu_t^{(i)} - \eta \cdot \frac{\partial F}{\partial \mu_t^{(i)}} \quad \text{where} \quad \frac{\partial F}{\partial \mu_t^{(i)}} \approx \varepsilon_t^{(i)} - \left( \frac{\partial f^{(i+1)}}{\partial \mu_t^{(i)}} \right)^T \varepsilon_t^{(i+1)} \tag{15}$$

Here, $\eta$ is a learning rate. This update rule adjusts the model's beliefs ($\mu_t^{(i)}$) to reduce the prediction error it reports ($\varepsilon_t^{(i)}$) while also striving to better predict the level below (the second term).

Crucially, this formulation provides a direct computational basis for the terms in Eq. 1 from the main text. The sum of squared precision-weighted prediction errors, $\sum_i (\varepsilon_t^{(i)})^T (\Pi^{(i)})^{-1} \varepsilon_t^{(i)}$, corresponds to the **Inaccuracy** term. The degree to which the posterior beliefs ($\mu_t^{(i)}$) must be updated away from their prior predictions ($f^{(i)}(\dots)$) corresponds to the **Complexity** term.

## A.3 MATHEMATICAL JUSTIFICATION OF THE FREE ENERGY PROXY

Our theoretical framework posits that aesthetic experience is linked to the dynamics of variational free energy minimization. The formal definition of free energy (Eq. 1) is defined over the latent causes $\mathbf{v}$ of sensory signals $\mathbf{s}$. However, for computational tractability within the APML architecture, we introduce a proxy, $\mathcal{F}_t$, that operates on the belief space of the Knowledge Backbone, as represented by the context vectors $\mathbf{c}_t$. This section provides the mathematical justification for treating $\mathcal{F}_t = \mathcal{I}_t + \mathcal{K}_t$ as a principled and valid proxy for the true variational free energy.

The core of our argument is that the `FEP-RNN` does not model the raw sensory world directly, but instead models the *dynamics of the beliefs* of another model (the Knowledge Backbone) that is observing the world. Therefore, the context vector $\mathbf{c}_t$ acts as a **sufficient statistic** for the sensory information and the backbone's internal state at time $t$. The free energy principle can thus be applied to this higher-level state space.

### A.3.1 JUSTIFICATION OF THE INACCURACY TERM PROXY ($\mathcal{I}_t$)

The theoretical **Inaccuracy** term is the expected negative log-likelihood of the sensory data, given the model's beliefs: $\mathbb{E}_{q(\mathbf{v}|\mu)}[-\ln P(\mathbf{s}|\mathbf{v}, \mathcal{M})]$. It quantifies the model's prediction error, or "surprisal," at the sensory level.

In our implementation, the proxy for this term is the surprisal of the *current context vector* given the history of past context vectors:

$$\mathcal{I}_t(\mathbf{c}_t) = -\log p_{\theta_{FEP}}(\mathbf{c}_t | \mathbf{c}_{<t}) \tag{16}$$

The validity of this proxy rests on the function of the Adapter module and the nature of $\mathbf{c}_t$. The context vector $\mathbf{c}_t$ is engineered to be a compressed representation of the Knowledge Backbone's posterior belief after observing the sensory input up to step $t$. Therefore, a highly surprising context vector (high $\mathcal{I}_t$) implies that the underlying sensory data was also highly surprising to the Knowledge Backbone. The `FEP-RNN`'s task is to predict the evolution of these beliefs. Its failure to do so, quantified by $\mathcal{I}_t$, serves as a direct, computable measure of the novelty or unpredictability of the musical stream as perceived by the primary generative model.

### A.3.2 Justification of the Complexity Term Proxy ($\mathcal{K}_t$)

The theoretical **Complexity** term is the Kullback-Leibler (KL) divergence between the posterior and prior beliefs over latent causes: $D_{\mathrm{KL}}[q(\mathbf{v}|\mu)||P(\mathbf{v}|\mathcal{M})]$. This term quantifies the "cost" of belief updating—the amount of information gained upon observing new data, which pushes the posterior away from the prior.

Our proxy for this term is a direct instantiation of this principle in the belief space of the FEP-RNN:

$$\mathcal{K}_t(\mathbf{c}_t) = D_{\mathrm{KL}}\left[q_{\theta_{FEP}}(\mathbf{c}_t|\mathbf{c}_{\leq t})\,||\,p_{\theta_{FEP}}(\mathbf{c}_t|\mathbf{c}_{<t})\right] \tag{17}$$

Here, the terms have precise analogues to the formal definition:

- **The Prior** ($p_{\theta_{FEP}}(\mathbf{c}_t|\mathbf{c}_{<t})$)**:** This is the predictive distribution for the context vector at time $t$, formulated by the FEP-RNN based on all information available *before* the observation at time $t$. It represents the model's expectation.
- **The Posterior** ($q_{\theta_{FEP}}(\mathbf{c}_t|\mathbf{c}_{\leq t})$)**:** This is the approximate posterior distribution over the context vector at time $t$, formulated *after* incorporating the new observation $\mathbf{c}_t$. It represents the model's updated belief.

Thus, $\mathcal{K}_t$ is not merely an analogy but a direct and principled implementation of the KL-divergence cost of belief updating, applied to the abstract state-space representation that the FEP-RNN operates on.

In summary, by treating the Knowledge Backbone's belief dynamics as the "sensory data" for the FEP-RNN, our proxy $\mathcal{F}_t = \mathcal{I}_t + \mathcal{K}_t$ becomes a tractable and theoretically-grounded instantiation of the variational free energy.

### A.4 Justification of the Aesthetic Reward as the Rate of Free Energy Reduction

The central thesis of our work, formalized in Eq. 2, is that aesthetic pleasure does not correlate with a state of low prediction error, but rather with the *process* of successfully reducing that error. This section provides a mathematical argument for why maximizing the aesthetic reward, $A_t \triangleq -dF_t/dt$, is a more effective objective for learning complex and engaging generative processes than directly minimizing free energy itself.

### A.4.1 Discretization and Interpretation as Learning Progress

In a computational model that operates in discrete time steps, the continuous temporal derivative $-dF_t/dt$ is approximated by the finite difference from the previous time step. The aesthetic reward at step $t$ is thus:

$$A_t \approx \mathcal{F}_{t-1} - \mathcal{F}_t \tag{18}$$

where $\mathcal{F}_t$ is our tractable free energy proxy. The cumulative reward over a sequence is $\sum_t A_t = \mathcal{F}_{\mathrm{initial}} - \mathcal{F}_{\mathrm{final}}$, which is precisely the total reduction in free energy, our CAV metric.

Maximizing $A_t$ at each step encourages the generative agent to select actions (i.e., generate tokens) that lead to the greatest possible reduction in its own uncertainty and predictive error. This objective can be interpreted as maximizing the agent's **epistemic progress** or **rate of learning**. Let's consider the alternatives:

- **Minimizing $\mathcal{F}_t$ directly:** An agent with this objective would seek out the most predictable states possible to minimize surprisal and complexity. In a musical context, this leads to generating silence or a monotonous, repeating note—a state of minimal free energy, but also minimal aesthetic value (boredom).
- **Maximizing $\mathcal{F}_t$:** This would lead the agent to seek out maximally surprising and complex states, resulting in the generation of chaotic, unpredictable noise. This corresponds to an aversive aesthetic experience.

Our objective, maximizing $A_t = \mathcal{F}_{t-1} - \mathcal{F}_t$, forces a dynamic and fruitful balance. To achieve a high reward, the agent must first be in a state of relatively high uncertainty ($\mathcal{F}_{t-1}$ is large), and then successfully transition to a state of lower uncertainty ($\mathcal{F}_t$ is small). This incentivizes the creation of "learnable surprises"—sequences that are neither trivially predictable nor intractably chaotic, but instead offer sustained opportunities for the model to successfully update its beliefs and reduce its prediction error.

A.4.2 CONNECTION TO THE META-LEARNING OBJECTIVE

This principle forms the mathematical foundation for our choice of the meta-learning objective in Eq. 3. The meta-learning process can be viewed as optimizing an agent's intrinsic capacity for efficient learning across diverse environments (musical styles).

- **The Inner Loop:** The fast adaptation performed in the inner loop on a support set $\mathcal{D}_{\text{supp}}^{(i)}$ is a direct simulation of aesthetic engagement with a new style. Minimizing the free energy loss, $\mathcal{L}_{\text{FreeEnergy}}$, over the support set is equivalent to maximizing the cumulative aesthetic reward $A_t$ within that style. The model is rewarded for how quickly and effectively it can reduce its uncertainty about the new style's conventions. An agent that can achieve a large reduction in $\mathcal{F}$ over a few examples is, by definition, an efficient few-shot learner.
- **The Outer Loop:** The outer loop updates the aesthetic priors ($\theta_{FEP,0}$) to minimize the expected free energy on the query set *after* this fast adaptation has occurred. Mathematically, this means the outer loop is optimizing the parameters $\theta_{FEP,0}$ to serve as a better inductive bias, such that the inner-loop learning process (the minimization of free energy) becomes more effective and generalizable.

Therefore, optimizing for the rate of free energy reduction is not an arbitrary choice; it is a principled objective that directly quantifies learning efficiency. By embedding this objective within a meta-learning framework, we train an agent that is not optimized to simply know musical styles, but is optimized to *enjoy the process of learning them*.

A.5 AESTHETIC DYNAMICS: DUAL TIMESCALE LEARNING

The process of engaging with music and forming aesthetic judgments unfolds over two distinct timescales. Our framework captures this by distinguishing between the rapid, real-time process of perceptual inference and the slower, experience-dependent process of model learning. This dual-timescale dynamic provides a principled, neuro-inspired foundation for the meta-learning architecture of APML.

**Fast Dynamics: Perceptual Inference (The Inner Loop)**     This process operates on a millisecond-to-second timescale, corresponding to the act of listening to a piece of music in real time. The objective during this phase is to minimize the instantaneous free energy $F_t$ by rapidly adjusting the neural activity of the representation units, $\mu$, while the model's synaptic weights, $\theta$, remain fixed.

$$\mu^* = \arg \min_{\mu} F_t(\mu|\mathbf{s}; \theta) \tag{19}$$

This is achieved through the predictive coding update rules described in Eqs. 14 and 15. This fast optimization process is the computational correlate of perception and understanding.

Critically, this fast dynamic corresponds directly to the **inner loop** of our meta-learning algorithm. When APML is presented with a few examples of a novel musical style (a new task), it performs this rapid free-energy minimization to form a task-specific posterior representation. The efficiency of this process—quantified by the rate of free energy reduction, $-dF_t/dt$—is the embodiment of the aesthetic reward signal defined in our theory.

**Slow Dynamics: Model Learning (The Outer Loop)**     This process operates over longer timescales, reflecting the gradual changes that occur across many listening experiences. The objective here is to minimize the long-term average of free energy, $\langle F \rangle_t$, by adjusting the model's parameters—the synaptic weights $\theta$ and precisions $\Pi$. This corresponds to learning the statistical regularities of a musical environment and is a form of error-gated Hebbian plasticity. The update rule for the weights follows the gradient of the average free energy (Friston, 2009; Clark, 2013):

$$\Delta \theta \propto -\frac{\partial \langle F \rangle_t}{\partial \theta} \approx \left\langle \sum_i \left( (\varepsilon_t^{(i)})^T \Pi^{(i)} \frac{\partial g^{(i)}}{\partial \theta} + (\mu_t^{(i)} - f^{(i)})^T \Pi^{(i)} \frac{\partial f^{(i)}}{\partial \theta} \right) \right\rangle_t \tag{20}$$

This rule ensures that synaptic connections are modified only to the extent that they can reduce long-term, average prediction error. Similarly, the precision parameters are updated to reflect the reliability of predictions at each level of the hierarchy.

This slow learning process is the direct analogue of the **outer loop** in our meta-learning framework. By optimizing the model parameters ($\theta_{FEP,0}$) based on the performance of the inner loop across many different musical styles (tasks), the model is not just learning one style, but is learning a set of universal priors that make future inference more efficient. This formalizes the notion of

"taste formation" as a meta-learning problem aimed at optimizing the capacity for rapid aesthetic appreciation (i.e., efficient free energy minimization) in any new context (Luppi et al., 2024; Hosp et al., 2021).

### A.6 QUANTIFYING MUSICAL PHENOMENA: THE DYNAMICAL FINGERPRINTS OF AESTHETICS

The strength of the active inference framework lies in its ability to translate subjective aesthetic experiences into objective, quantifiable dynamics within the generative model. By analyzing the interplay of prediction and belief updating across the hierarchy, we can identify unique "fingerprints" for different musical phenomena, directly substantiating the conceptual claims in Figure 1. To do this, we define two key metrics at each level $i$ of the model:

- **Hierarchical Prediction Error (HPE):** $\mathcal{E}_t^{(i)} = \|\varepsilon_t^{(i)}\|^2$. This metric quantifies the energy of the precision-weighted prediction error at level $i$, corresponding to the instantaneous sensory surprisal or *inaccuracy*.
- **Hierarchical Belief Update (HBU):** $\mathcal{U}_t^{(i)} = D_{\mathrm{KL}}[q(\mathbf{v}_t^{(i)})\|p(\mathbf{v}_t^{(i)})]$. This is the Kullback-Leibler divergence between the posterior belief $q(\mathbf{v}_t^{(i)})$ after observing the data, and the prior belief $p(\mathbf{v}_t^{(i)})$. It quantifies the amount of information gained, corresponding to the *complexity* term of free energy.

The temporal evolution of these metrics across the hierarchy forms the basis for our analysis:

**Monotonous vs. Chaotic Sequences**   These two extremes of aesthetic experience are clearly distinguished by their HPE dynamics.

- A **monotonous** sequence (e.g., a repeating single note) is highly predictable. After a brief initial phase, the model's predictions become perfect. Consequently, $\forall i, t : \mathcal{E}_t^{(i)} \to 0$ and $\mathcal{U}_t^{(i)} \to 0$. The free energy is minimal and stable, causing its temporal gradient, our proxy for aesthetic reward, to be zero ($dF/dt \approx 0$). This is the signature of boredom.
- A **chaotic** sequence (e.g., white noise) is, by definition, unpredictable. The model consistently fails to reduce its prediction error. Consequently, $\forall i : \langle \mathcal{E}_t^{(i)} \rangle_t \gg 0$, and the model is unable to systematically lower its free energy. The average gradient is non-negative ($\langle dF/dt \rangle_t \geq 0$), corresponding to a confusing or aversive experience.

**"Catchy" Music: The Mid-Level Sweet Spot**   Aesthetically pleasing and accessible music, such as a catchy pop song, often provides a "sweet spot" of learnable surprise concentrated at a specific hierarchical level. Its dynamical fingerprint is characterized by:

- **High-Level Predictability:** The highest level ($L_3$) exhibits low prediction error ($\mathcal{E}_t^{(3)} \approx 0$) due to conventional structures like verse-chorus forms. This structural simplicity reduces the overall cognitive load.
- **Mid-Level Engagement:** The primary source of aesthetic reward comes from the musical structure level ($L_2$). Here, the model is engaged in a continuous and successful process of predicting and updating beliefs about melody and harmony. This results in a sustained, negative free energy gradient ($\langle dF_t^{(2)}/dt \rangle_t < 0$) driven by resolvable "surprises" in the melodic and harmonic contours.

**"Profound" Music: The Power of Cross-Level Causality**   The experience of "profound" or deeply moving music (e.g., a complex symphony) involves a more sophisticated dynamic that spans the entire hierarchy. Its fingerprint is defined by **cross-level causality**, where a successful belief update at a high level of abstraction precipitates a cascade of simplified prediction tasks at lower levels.

- **The "Aha!" Moment:** This experience is initiated by a significant belief update at the highest level ($\mathcal{U}_t^{(3)} \gg 0$), corresponding to grasping a complex formal or thematic idea.
- **Causal Reduction of Uncertainty:** This high-level insight provides a powerful top-down prior that dramatically simplifies the prediction task at lower levels. The moment the overarching structure is understood, seemingly complex melodic or harmonic details become predictable.

This causal relationship can be formally quantified using techniques like the Granger Causality test (Park et al., 2024; Stokes et al., 2022). The Granger Causality from a higher level $L$ to a lower level $l$ can be measured by the degree to which the history of the high-level states improves the prediction of the low-level error:

$$GC_{L\to l} = \ln \frac{\text{Var}(\mathcal{E}_t^{(l)}|\text{history of } \mathcal{E}^{(l)})}{\text{Var}(\mathcal{E}_t^{(l)}|\text{history of } \mathcal{E}^{(l)} \text{ and } \mu^{(L)})} \tag{21}$$

A significantly positive $GC_{L\to l}$ value provides quantitative evidence that the high-level representation is causally simplifying the low-level perceptual task. A profound aesthetic experience is therefore characterized by a high Granger Causality score, where the successful resolution of high-level complexity leads to a highly efficient, system-wide reduction in free energy, yielding a large and sustained aesthetic reward.

## B APPENDIX: IMPLEMENTATION DETAILS OF THE APML ARCHITECTURE

This appendix provides a detailed specification of the key components of the Aesthetic Priors Meta-Learner (APML) described in Section 4. We elaborate on the engineering solutions and mathematical formulations chosen to ensure that the model is not only computationally efficient but also faithful to the principles of our active inference framework.

### B.1 THE ACTIVE INFERENCE INTERFACE: FROM BELIEF EXTRACTION TO LOGIT-SPACE GUIDANCE

The interface between the Knowledge Backbone (`MuseNet-XL`) and the Aesthetic Core (`FEP-RNN`) is the functional heart of APML. Its implementation is critical for effective and efficient communication between the two modules.

#### B.1.1 CONTEXT VECTOR FORMULATION VIA ATTENTION-POOLED STATE EXTRACTION

The context vector $\mathbf{c}_t$ serves as a low-dimensional summary of the Knowledge Backbone's high-dimensional belief state. To create a representation that captures both semantic content and predictive uncertainty, we adopt an approach based on attention pooling (Vaswani et al., 2017; Zha et al., 2025).

At each generation step $t$, we extract two signals from the final layer of `MuseNet-XL`:

1. **Semantic State ($h_t^{(L)}$):** The sequence of hidden state vectors from the top Transformer layer.
2. **Attention Distribution ($A_t^{(L)}$):** The final self-attention probability matrix.

The semantic part of the context vector is formed by pooling the hidden states, weighted by their final attention scores. The uncertainty is quantified by the entropy of this attention distribution. Formally, given the attention weights $\alpha_t$ derived from $A_t^{(L)}$, the components of the context vector $\mathbf{c}_t$ are:

$$\mathbf{c}_{t,\text{semantic}} = \sum_{i=1}^{T} \alpha_{t,i} h_{t,i}^{(L)} \tag{22}$$

$$c_{t,\text{uncertainty}} = -\sum_{i=1}^{T} \alpha_{t,i} \log(\alpha_{t,i} + \epsilon) \tag{23}$$

where $T$ is the context sequence length and $\epsilon$ is a small constant for numerical stability. The final context vector is the concatenation of these two components, passed through the learned Adapter module: $\mathbf{c}_t = \text{MLP}_{\phi_{adapt}}([\mathbf{c}_{t,\text{semantic}}; c_{t,\text{uncertainty}}])$. This method provides a computationally efficient ($O(T)$) and empirically validated way to extract a rich belief state.

#### B.1.2 PDPM VIA EFFICIENT LOGIT-SPACE GUIDANCE

The Probability Distribution Potential Modulation (PDPM) mechanism, conceptually described as a KL-divergence based modulation, is computationally intractable for large vocabularies ($O(|\mathcal{V}|)$). We therefore implement it using a highly efficient logit-space manipulation technique inspired by Classifier-Free Guidance (CFG) (Ho et al., 2021; Saharia et al., 2022).

This approach reframes the guidance signal $\mathbf{g}_t$ not as a potential cost, but as a directional vector in the logit space. The `FEP-RNN` outputs an aesthetic guidance embedding $\mathbf{e}_t$, which is projected to the vocabulary dimension. This projected vector directly modulates the original logits $\mathbf{o}_t$ from `MuseNet-XL`. The final, aesthetically-guided logits $\mathbf{o}'_t$ are computed as:

$$\mathbf{o}'_t = \mathbf{o}_t + \gamma \cdot (\mathbf{W}_g \mathbf{e}_t) \tag{24}$$

where $\mathbf{W}_g$ is a learnable projection matrix and $\gamma$ is a scalar hyperparameter controlling the guidance strength. Note that in a typical CFG formulation, the guidance term is $(\mathbf{o}_{\text{cond}} - \mathbf{o}_{\text{uncond}})$. Our formulation simplifies this, treating the projected aesthetic embedding as the directional guidance 'a priori'. This method achieves the same conceptual goal—steering the output distribution—with a minimal computational overhead of $O(d_{\text{embed}})$, where $d_{\text{embed}} \ll |\mathcal{V}|$.

### B.1.3 MATHEMATICAL JUSTIFICATION OF PDPM AS A GRADIENT-BASED STEERING MECHANISM

The PDPM mechanism, implemented as logit-space guidance (Eq. 24), provides an efficient way to steer the generation process. Here, we provide a mathematical justification to show that this mechanism is not ad-hoc, but is in fact a first-order approximation of a more principled objective: selecting the next token $x_{t+1}$ that maximizes the anticipated aesthetic reward, $A_{t+1}$.

Our goal at time step $t$ is to choose the next token $x_{t+1}$ to maximize the expected immediate aesthetic reward, $A_{t+1} \approx \mathcal{F}_t - \mathcal{F}_{t+1}$. Since $\mathcal{F}_t$ is constant with respect to the choice of $x_{t+1}$, this is equivalent to minimizing the expected free energy of the *next* state, $\mathcal{F}_{t+1}$:

$$x^*_{t+1} = \arg\min_{v \in \mathcal{V}} \mathbb{E}\left[\mathcal{F}_{t+1} | \mathbf{x}_{\leq t}, x_{t+1} = v\right] \tag{25}$$

where $\mathcal{V}$ is the vocabulary of all possible tokens. Directly computing this expectation for every token $v$ is intractable, as it would require simulating the Knowledge Backbone's state transition for each candidate token. Our PDPM mechanism serves as a computationally feasible approximation of this optimization.

The free energy at the next step, $\mathcal{F}_{t+1}$, is a function of the next context vector $\mathbf{c}_{t+1}$. We can approximate the objective in Eq. 25 by taking a first-order Taylor expansion of $\mathcal{F}_{t+1}$ around the current state's context vector, $\mathbf{c}_t$. However, a more direct approach is to consider the components of $\mathcal{F}_{t+1} = \mathcal{I}_{t+1} + \mathcal{K}_{t+1}$:

$$\mathcal{F}_{t+1}(\mathbf{c}_{t+1}) = -\log p_{\theta_{FEP}}(\mathbf{c}_{t+1}|\mathbf{c}_{\leq t}) + D_{\text{KL}}\left[q_{\theta_{FEP}}(\mathbf{c}_{t+1}|\mathbf{c}_{\leq t+1}) \,\|\, p_{\theta_{FEP}}(\mathbf{c}_{t+1}|\mathbf{c}_{\leq t})\right] \tag{26}$$

The dominant term in this expression for determining the "surprise" of the next state is the inaccuracy or negative log-likelihood, $-\log p_{\theta_{FEP}}(\mathbf{c}_{t+1}|\mathbf{c}_{\leq t})$. The complexity term $\mathcal{K}_{t+1}$ quantifies the subsequent belief update, which is harder to predict. Therefore, a reasonable and common approximation in active inference is to primarily focus on minimizing the expected future inaccuracy (also known as resolving uncertainty). Our objective simplifies to:

$$x^*_{t+1} \approx \arg\min_{v \in \mathcal{V}} \mathbb{E}\left[-\log p_{\theta_{FEP}}(\mathbf{c}_{t+1}|\mathbf{c}_{\leq t})\right] \quad \text{where } \mathbf{c}_{t+1} \text{ depends on } x_{t+1} = v \tag{27}$$

This is equivalent to finding the token $v$ that leads to a next context state $\mathbf{c}_{t+1|v}$ that is most probable under the `FEP-RNN`'s current predictive model. In other words, the agent is guided to generate tokens that conform to its learned model of "aesthetically pleasing" belief dynamics.

Now, let's connect this to our logit-space guidance. The `FEP-RNN` is trained to predict the next context vector. The aesthetic guidance embedding $\mathbf{e}_t$ can be interpreted as an encoding of the desired "target" for the next state, $\mathbf{c}_{t+1}$. The logit-space guidance signal is $\mathbf{g}_t = \mathbf{W}_g \mathbf{e}_t$. The final probability of selecting a token $v$ is proportional to:

$$P(x_{t+1} = v|\mathbf{x}_{\leq t}) \propto \exp(o_t(v) + \gamma \cdot g_t(v)) \tag{28}$$

where $o_t(v)$ is the original logit from the Knowledge Backbone for token $v$. The term $g_t(v)$ acts as a reward or "potential" for choosing token $v$. If the guidance signal $\mathbf{g}_t$ is structured such that tokens leading to a lower future free energy receive a higher value, then this modulation directly pushes the sampling distribution towards the objective.

The final step is to argue that the training process of the `FEP-RNN` shapes the guidance signal $\mathbf{g}_t$ to be precisely this potential. The `FEP-RNN` is trained via the free energy loss, which rewards the model for making accurate predictions about the evolution of context vectors. Consequently, the embedding $\mathbf{e}_t$ that it learns to produce represents an "informed guess" about a future state $\mathbf{c}_{t+1}$ that would be easy for it to assimilate (i.e., a state that would result in a low $\mathcal{F}_{t+1}$). The projection $\mathbf{W}_g$ maps this abstract "desirable state" into a concrete set of rewards over the vocabulary. There-

fore, the PDPM mechanism effectively adds a learned, gradient-based bias to the sampling process, steering the generation towards trajectories that the aesthetic core predicts will maximize the rate of successful learning.

## B.2 COMPUTATIONAL FORMULATION OF THE FREE ENERGY PROXY

The instantaneous free energy proxy, $\mathcal{F}_t = \mathcal{I}_t + \mathcal{K}_t$, relies on the ability of the `FEP-RNN` to form probabilistic predictions and requires a stable, differentiable method for computing the KL divergence between its prior and posterior beliefs.

### B.2.1 THE FEP-RNN AS A HIERARCHICAL DYNAMIC GMM

As described in the main text, the `FEP-RNN` is a 3-layer GRU, where each layer is equipped with a Mixture Density Network (MDN) head. At each step $t$, the MDN head for layer $l$ outputs the parameters $\{\pi_{t,k}^{(l)}, \boldsymbol{\mu}_{t,k}^{(l)}, \boldsymbol{\Sigma}_{t,k}^{(l)}\}_{k=1}^K$ of a Gaussian Mixture Model (GMM). This GMM represents the model's belief about the *next* context vector $\mathbf{c}_{t+1}$. The overall prior predictive distribution, $p_{\theta_{FEP}}(\mathbf{c}_{t+1}|\mathbf{c}_{\leq t})$, is a hierarchical mixture of these GMMs, allowing for complex, multi-modal predictions.

### B.2.2 STABLE KL DIVERGENCE APPROXIMATION FOR COMPLEXITY

The complexity term, $\mathcal{K}_t = D_{\mathrm{KL}}[q_{\theta_{FEP}}(\mathbf{c}_{t+1}|\mathbf{c}_{\leq t+1}) \,||\, p_{\theta_{FEP}}(\mathbf{c}_{t+1}|\mathbf{c}_{<t+1})]$, involves computing the KL divergence between two GMMs (the posterior $q$ and the prior $p$). This quantity has no closed-form solution and can be numerically unstable to compute directly.

We employ a standard and robust technique from variational inference: **Monte Carlo approximation with reparameterized sampling** (Kingma & Welling, 2013). The KL divergence is estimated by drawing samples from the posterior distribution $q$ and evaluating the log-density ratio under both distributions.

$$D_{\mathrm{KL}}[q||p] \approx \frac{1}{N} \sum_{n=1}^{N} [\log q(\mathbf{z}_n) - \log p(\mathbf{z}_n)], \quad \text{where } \mathbf{z}_n \sim q(\cdot) \tag{29}$$

To ensure this process is differentiable, the samples $\mathbf{z}_n$ are drawn using the reparameterization trick: a sample from a GMM is generated by first sampling a mixture component, then sampling from the corresponding Gaussian via $\mathbf{z}_n = \boldsymbol{\mu}_k + \mathbf{L}_k \boldsymbol{\epsilon}_n$, where $\mathbf{L}_k$ is the Cholesky decomposition of the covariance matrix $\boldsymbol{\Sigma}_k$ and $\boldsymbol{\epsilon}_n \sim \mathcal{N}(0, \mathbf{I})$.

The log-densities $\log q(\mathbf{z}_n)$ and $\log p(\mathbf{z}_n)$ are computed using the log-sum-exp trick for numerical stability. This method provides a low-variance, unbiased, and fully differentiable estimate of the complexity term, which is essential for end-to-end training of the `FEP-RNN`.

## C APPENDIX: ALGORITHMS AND TRAINING PROCEDURES

This section provides detailed algorithmic descriptions for the training and generation processes of the Aesthetic Priors Meta-Learner (APML), complementing the architectural details in Appendix B.

### C.1 THE THREE-PHASE TRAINING ALGORITHM

The complete training of APML is structured into three distinct phases, as outlined in Algorithm 1. This procedure is designed to first build a robust foundation of musical knowledge, then establish a collaborative interface between the two cores, and finally, instill the adaptive aesthetic priors through meta-learning.

### C.2 AESTHETICALLY GUIDED AUTOREGRESSIVE GENERATION

During inference, APML generates music token-by-token. At each step, the Knowledge Backbone proposes a distribution over the next token, which is then modulated by a guidance signal from the Aesthetic Core before the final token is sampled. This process, detailed in Algorithm 2, realizes the Probability Distribution Potential Modulation (PDPM) mechanism through efficient logit-space manipulation, as described in Appendix B.1.

---

**Algorithm 1** APML Three-Phase Training Procedure

---

1: **Input:** Knowledge Backbone $\theta_{XL}$; Aesthetic Core $\theta_{FEP}$; Adapter $\phi_{adapt}$
2: **Input:** Pre-training data $\mathcal{D}_{\text{pretrain}}$; Adaptation data $\mathcal{D}_{\text{adapt}}$; Meta-learning tasks $p(\mathcal{T})$
3: **Input:** Hyperparameters: $\lambda_F$, meta-learning rates $\alpha, \beta$, inner update steps $k$

    **Phase 1: Knowledge Foundation Pre-training**
4: Initialize $\theta_{XL}$ with pre-trained weights or randomly.
5: Train $\theta_{XL}$ on $\mathcal{D}_{\text{pretrain}}$ by minimizing the Negative Log-Likelihood (NLL) loss:
6: $\theta_{XL}^* \leftarrow \arg\min_{\theta_{XL}} \mathbb{E}_{\mathbf{x} \sim \mathcal{D}_{\text{pretrain}}} [\mathcal{L}_{\text{NLL}}(\theta_{XL}; \mathbf{x})]$

    **Phase 2: Interface Adaptation Fine-tuning**
7: Freeze the majority of parameters in the optimized backbone $\theta_{XL}^*$.
8: Initialize $\theta_{FEP}$ and $\phi_{adapt}$ randomly.
9: Jointly train $\theta_{FEP}$ and $\phi_{adapt}$ (and unfrozen parts of $\theta_{XL}^*$) on $\mathcal{D}_{\text{adapt}}$.
10: The objective is a multi-task loss combining NLL and the free energy proxy $\mathcal{F}_t$:
11: $\theta_{FEP}^*, \phi_{adapt}^* \leftarrow \arg\min_{\theta_{FEP}, \phi_{adapt}} \mathbb{E}_{\mathbf{x} \sim \mathcal{D}_{\text{adapt}}} [\mathcal{L}_{\text{NLL}} + \lambda_F \cdot \mathcal{L}_{\text{FreeEnergy}}]$
12: where $\mathcal{L}_{\text{FreeEnergy}} = \frac{1}{|\mathbf{x}|} \sum_t \mathcal{F}_t(\theta_{FEP}, \phi_{adapt}; \mathbf{x}_{\leq t})$

    **Phase 3: Aesthetic Priors Meta-Learning**
13: Initialize the Aesthetic Core with the adapted weights $\theta_{FEP,0} \leftarrow \theta_{FEP}^*$.
14: **for** each meta-iteration **do**
15:     Sample a batch of tasks $\mathcal{T}_i \sim p(\mathcal{T})$, where each $\mathcal{T}_i = \{\mathcal{D}_{\text{supp}}^{(i)}, \mathcal{D}_{\text{query}}^{(i)}\}$.
16:     Initialize meta-gradient $\nabla_{\theta_{FEP,0}}\mathcal{L} \leftarrow 0$.
17:     **for** each task $\mathcal{T}_i$ **do**
18:         // *Inner Loop: Fast Adaptation*
19:         Create a task-specific copy of the core: $\theta_{FEP,i}' \leftarrow \theta_{FEP,0}$.
20:         **for** $j = 1, \ldots, k$ **do**
21:             Compute free energy loss on the support set: $L_{\text{supp}} = \mathcal{L}_{\text{FreeEnergy}}(\theta_{FEP,i}'; \mathcal{D}_{\text{supp}}^{(i)})$.
22:             Update task-specific parameters: $\theta_{FEP,i}' \leftarrow \theta_{FEP,i}' - \alpha\nabla_{\theta'}L_{\text{supp}}$.
23:         **end for**
24:         // *Evaluate on Query Set for Meta-Update*
25:         Compute free energy loss on the query set: $L_{\text{query}} = \mathcal{L}_{\text{FreeEnergy}}(\theta_{FEP,i}'; \mathcal{D}_{\text{query}}^{(i)})$.
26:         Accumulate meta-gradient: $\nabla_{\theta_{FEP,0}}\mathcal{L} \leftarrow \nabla_{\theta_{FEP,0}}\mathcal{L} + \nabla_{\theta_{FEP,0}}L_{\text{query}}$.
27:     **end for**
28:     // *Outer Loop: Update Aesthetic Priors*
29:     $\theta_{FEP,0} \leftarrow \theta_{FEP,0} - \beta \cdot \nabla_{\theta_{FEP,0}}\mathcal{L}$.
30: **end for**

31: **Return:** The fully trained APML system composed of $\theta_{XL}^*$, $\phi_{adapt}^*$, and the meta-learned aesthetic priors $\theta_{FEP,0}$.

---

## C.3 FROM PREDICTIVE CODING TO RECURRENT NETWORKS: A PRACTICAL IMPLEMENTATION

A key aspect of our framework is the concrete link between the neuro-inspired theory of Hierarchical Predictive Coding (HPC) and the modern deep learning architecture of our `FEP-RNN`. A multi-layer Gated Recurrent Unit (GRU) network serves as a powerful and computationally efficient implementation of the core HPC principles (Millidge et al., 2020; Whittington et al., 2024). The correspondence is as follows:

- **Hierarchical Structure:** The distinct layers of the GRU ($l = 1, 2, 3$) directly correspond to the levels in the HPC hierarchy ($L_1, L_2, L_3$). Lower layers model fast, sensory-level dynamics, while higher layers capture slower, more abstract contextual information, mirroring the multi-scale nature of musical structure.
- **Top-Down Predictions:** In HPC, higher levels provide contextual priors to lower levels (Eq. 13). In a multi-layer GRU, the hidden state of the layer above, $\mathbf{h}_t^{(l)}$, is passed as input to the layer below, $\mathbf{h}_t^{(l-1)}$, at each time step. This directly implements the top-down predictive guidance mechanism: $\mathbf{h}_t^{(l-1)} = \text{GRU}(\mathbf{h}_{t-1}^{(l-1)}, \mathbf{h}_t^{(l)})$.

---

**Algorithm 2** Aesthetically Guided Autoregressive Generation

---

1: **Input:** Trained APML model $(\theta_{XL}, \theta_{FEP}, \phi_{adapt})$, prompt sequence $\mathbf{x}_{\text{prompt}}$, generation length $L_{gen}$
2: **Input:** Guidance strength hyperparameter $\gamma$
3: Initialize the generated sequence: $\mathbf{x} \leftarrow \mathbf{x}_{\text{prompt}}$.
4: **for** $t = |\mathbf{x}_{\text{prompt}}|, \ldots, L_{gen} - 1$ **do**
5:     // *Step 1: Knowledge Backbone Prediction*
6:     Obtain original output logits from `MuseNet-XL`: $\mathbf{o}_t \in \mathbb{R}^{|\mathcal{V}|} \leftarrow \text{Forward}_{XL}(\mathbf{x}_{\leq t})$.
7:     // *Step 2: Belief State Extraction*
8:     Extract the backbone's belief state (e.g., hidden states $h_t^{(L)}$, attention entropy $E_t$).
9:     Compute the context vector via the Adapter: $\mathbf{c}_t \leftarrow \text{Adapter}_{\phi_{adapt}}(h_t^{(L)}, E_t)$.
10:     // *Step 3: Aesthetic Core Guidance Signal*
11:     Update the `FEP-RNN` state with the sequence of context vectors $\mathbf{c}_{\leq t}$.
12:     Generate an aesthetic guidance embedding: $\mathbf{e}_t \in \mathbb{R}^{d_{embed}} \leftarrow \text{Forward}_{FEP}(\mathbf{c}_{\leq t})$.
13:     // *Step 4: Logit-Space Modulation (PDPM Implementation)*
14:     Project the guidance embedding to the vocabulary dimension: $\mathbf{g}_t \leftarrow \mathbf{W}_g \mathbf{e}_t$.
15:     Modulate the original logits: $\mathbf{o}_t' \leftarrow \mathbf{o}_t + \gamma \cdot \mathbf{g}_t$.
16:     // *Step 5: Sampling*
17:     Compute the final probability distribution: $P(\cdot | \mathbf{x}_{\leq t}) \leftarrow \text{softmax}(\mathbf{o}_t')$.
18:     Sample the next token: $x_{t+1} \sim P(\cdot | \mathbf{x}_{\leq t})$.
19:     Append the new token to the sequence: $\mathbf{x} \leftarrow \text{concat}(\mathbf{x}, x_{t+1})$.
20: **end for**
21: **Return:** The complete generated sequence $\mathbf{x}$.

---

- **Belief Updating and Error Correction:** The core GRU update mechanism can be interpreted as an amortized form of free-energy minimization. The update gate, $\mathbf{z}_t$, arbitrates between maintaining the prior belief (the previous hidden state $\mathbf{h}_{t-1}$) and incorporating new evidence (the current input). The reset gate, $\mathbf{r}_t$, acts as a precision-weighting mechanism, determining how much of the prior belief is used to form the proposal for the new state. This gating dynamic allows the network to flexibly update its internal representations in response to prediction errors, analogous to the update dynamics of the representation units ($\mu$) in HPC (Eq. 15).

This mapping ensures that our `FEP-RNN` is not merely a black-box function approximator but a structured generative model that embodies the core computational principles of our theoretical framework.

### C.3.1 STABLE KL DIVERGENCE APPROXIMATION FOR COMPLEXITY

The complexity term, $\mathcal{K}_t = D_{\text{KL}}[q_{\theta_{FEP}}(\mathbf{c}_t | \mathbf{c}_{\leq t}) \, || \, p_{\theta_{FEP}}(\mathbf{c}_t | \mathbf{c}_{<t})]$, involves computing the KL divergence between two GMMs (the posterior $q$ and the prior $p$). This quantity has no closed-form solution because the logarithm of a sum prevents analytical integration. We therefore employ a robust Monte Carlo approximation (Kingma & Welling, 2013; Hershey & Olsen, 2007).

The KL divergence is estimated by drawing $N$ samples from the posterior distribution $q$ and evaluating the average log-density ratio:

$$D_{\text{KL}}[q||p] \approx \frac{1}{N} \sum_{n=1}^{N} \left[ \log q(\mathbf{z}_n) - \log p(\mathbf{z}_n) \right], \quad \text{where } \mathbf{z}_n \sim q(\cdot) \tag{30}$$

To ensure this process is differentiable for backpropagation, the samples $\mathbf{z}_n$ are drawn using the reparameterization trick. A sample from a GMM is generated by first sampling a mixture component $k$ according to the mixture weights $\pi_k$, then sampling from the corresponding Gaussian via $\mathbf{z}_n = \boldsymbol{\mu}_k + \mathbf{L}_k \boldsymbol{\epsilon}_n$, where $\mathbf{L}_k$ is the Cholesky decomposition of the covariance matrix $\boldsymbol{\Sigma}_k$ and $\boldsymbol{\epsilon}_n \sim \mathcal{N}(0, \mathbf{I})$.

The log-densities $\log q(\mathbf{z}_n)$ and $\log p(\mathbf{z}_n)$ for a GMM with $K$ components are computed using the log-sum-exp trick for numerical stability:

$$\log p(\mathbf{z}) = C + \log \left( \sum_{k=1}^{K} \exp \left( \log \pi_k + \log \mathcal{N}(\mathbf{z} | \boldsymbol{\mu}_k, \boldsymbol{\Sigma}_k) - C \right) \right) \tag{31}$$

where $C = \max_k(\log \pi_k + \log \mathcal{N}(\mathbf{z}|\boldsymbol{\mu}_k, \boldsymbol{\Sigma}_k))$. This method provides a low-variance, unbiased, and fully differentiable estimate of the complexity term, essential for the end-to-end training of the `FEP-RNN`.

# D  APPENDIX: BENCHMARK CONSTRUCTION AND EVALUATION METRICS

This section details the data sources used to construct our Olympus-Meta Pipeline and provides precise mathematical definitions for all evaluation metrics employed in our experiments, ensuring full transparency and reproducibility.

## D.1  DATA SOURCES FOR THE OLYMPUS-META PIPELINE

The construction of our meta-learning benchmark relies on a curated selection of large-scale, publicly available MIDI datasets. Each dataset provides unique stylistic diversity, contributing to the breadth and challenge of the benchmark. Table 4 summarizes the key characteristics of these sources.

Table 4: Public MIDI datasets used in the construction of the Olympus-Meta Pipeline.

| Dataset | Typical Use | Musical Style Coverage | Scale (approx.) | Reference |
|---|---|---|---|---|
| Lakh MIDI (LMD) | Pre-training, Style Transfer | Broad (Pop, Rock, Classical, Jazz) | ∼176k files | (Raffel, 2016) |
| GiantMIDI-Piano | Piano Generation | Classical (Composer-specific) | ∼11k files, 1,200h | (Kong et al., 2020) |
| MAESTRO | Performance Generation | Classical (Competition-level) | ∼1.3k files, 200h | (Hawthorne et al., 2019) |
| JAZZ-HARMONY-DB | Jazz Harmony Analysis | Jazz (Swing, Bebop) | ∼1k standards | (Granroth-Wilding, 2018) |
| Essen Folk Song | Folk Music Generation | European Folk | ∼8.5k songs | (Schaffrath, 1995) |

## D.2  DETAILED EVALUATION METRICS

Below are the formal definitions for all 11 metrics used to evaluate model performance in Section 5. The metrics are grouped into three categories for clarity.

**Predictive Accuracy Metrics.** These metrics evaluate the model's fundamental ability to predict musical sequences correctly.

- **Negative Log-Likelihood (NLL):** Measures the average per-token prediction error. For a sequence $\mathbf{x} = (x_1, \ldots, x_T)$, it is the average of the negative logarithm of the probability assigned to the true next token:

$$\text{NLL} = -\frac{1}{T}\sum_{t=1}^{T} \log P(x_t|\mathbf{x}_{<t}, \theta) \tag{32}$$

  A lower NLL indicates a more accurate statistical model.
- **Accuracy (ACC):** Measures the percentage of tokens where the model's most probable prediction (*argmax*) matches the ground truth token.

$$\text{ACC} = \frac{1}{T}\sum_{t=1}^{T} \mathbb{I}(\arg\max P(\cdot|\mathbf{x}_{<t}, \theta) = x_t) \tag{33}$$

where $\mathbb{I}(\cdot)$ is the indicator function. It directly measures the model's fluency.

**Stylistic Consistency and Musicality Metrics.** These metrics assess the quality of the generated music from established musicological and information-theoretic perspectives.

- **Fréchet MIDI Distance (FMD):** Measures the 2-Wasserstein distance between the distributions of generated and real musical sequences in a latent space. Embeddings are extracted using a pre-trained MusicBERT model, and the distance is computed on the mean and covariance of these embeddings: $d^2 = ||\boldsymbol{\mu}_R - \boldsymbol{\mu}_G||_2^2 + \text{Tr}(\boldsymbol{\Sigma}_R + \boldsymbol{\Sigma}_G - 2(\boldsymbol{\Sigma}_R\boldsymbol{\Sigma}_G)^{1/2})$.
- **Style Classifier Confidence (SCC):** Measures how confidently a pre-trained style classifier assigns a generated sequence to the target style. For a classifier $C$ and a generated sequence $x$, it is the average of the maximum softmax probability: $\text{SCC} = \mathbb{E}_{x \sim G}[\max_y P_C(y|x)]$.
- **Pitch Count (PC):** The number of unique MIDI pitch values present in a generated sequence. For a sequence with notes having pitches $\{p_1, \ldots, p_T\}$, $\text{PC} = |\text{unique}(\{p_i\})|$.
- **Rhythmic Entropy (RE):** The Shannon entropy of the distribution of quantized note durations in a sequence. For a set of unique durations $\{d_j\}$ with probabilities $p(d_j)$, the entropy is: $\text{RE} = -\sum_j p(d_j) \log_2 p(d_j)$.
- **Tonal Clarity (TC):** Quantifies the strength of an implied musical key using the Krumhansl-Schmuckler algorithm. It is the maximum correlation between the sequence's pitch-class histogram and 24 standard key profiles.
- **Chord Progression Score (CPS):** A proxy for harmonic coherence, calculated as the average negative log-likelihood of a sequence's chord progression under a pre-trained chord sequence model.
- **Self-BLEU:** Measures the diversity of the generated output. It is the average BLEU-n score computed between all pairs of sequences within the generated set. A lower score indicates higher diversity.

**Theory Validation Metrics.** These novel metrics are designed specifically to test the core hypotheses of our theoretical framework by quantifying the dynamics of the free energy proxy, $\mathcal{F}_t$, during generation.

- **Free Energy-Descent Rate (FEDR):** Measures the average rate of reduction of the free energy proxy over the generation process. This directly quantifies the "learnable surprise" central to our theory. For a generated sequence of length $L_{gen}$:

$$\text{FEDR} = \frac{1}{L_{gen}} \sum_{t=1}^{L_{gen}} (\mathcal{F}_{t-1} - \mathcal{F}_t) \tag{34}$$

A higher FEDR indicates a more efficient and sustained process of uncertainty reduction.
- **Cumulative Aesthetic Value (CAV):** Measures the total reduction in free energy over the entire generation process, proxying for the overall aesthetic engagement. It is the net change from the initial state to the final state:

$$\text{CAV} = \mathcal{F}_{\text{initial}} - \mathcal{F}_{\text{final}} \tag{35}$$

A higher CAV suggests that the generative process has successfully navigated from a state of high uncertainty to one of resolution, a hallmark of a compelling aesthetic arc.

# E APPENDIX: DETAILED EXPERIMENTAL SETUP

This section provides a comprehensive overview of the experimental setup, detailing the construction of our reproducible benchmark and the precise hyperparameter configurations used for training and evaluating the APML model.

## E.1 THE OLYMPUS-META PIPELINE CONSTRUCTION PROCESS

The Olympus-Meta Pipeline is our standardized framework for creating the few-shot music style transfer benchmark. The pipeline, inspired by best practices in music information retrieval and generative modeling (Huang et al., 2020; Raffel, 2016), is designed to be fully reproducible and to ensure stylistic integrity across all meta-learning tasks. The process is comprised of four main stages.

**Step 1: Data Sourcing and Filtering.** The process begins with the aggregation of MIDI files from the sources listed in Table 4. A rigorous filtering protocol is then applied to ensure data quality and consistency for the downstream piano-focused task:

- **Instrument Filtering:** Using the `pretty_midi` library, we retain only files that contain at least one piano track (MIDI program numbers 0-15). All dedicated percussion tracks (typically found on MIDI channel 10) are discarded to maintain a focus on melodic and harmonic content.
- **Sequence Length Constraints:** To avoid trivial or overly complex examples, we filter out all MIDI files that, after tokenization, result in fewer than 256 or more than 10,000 tokens. This ensures a consistent level of complexity within the dataset.
- **Quality Control:** Files exhibiting anomalous characteristics are removed. This includes MIDI files with an exceptionally high note density (defined as ¿100 notes per second) or those containing invalid MIDI events (e.g., negative time shifts or invalid pitch values), which often indicate data corruption.

This filtering stage yields a high-quality corpus of approximately 100,000 piano-centric MIDI files, which serves as the foundation for the benchmark.

**Step 2: Data Representation and Tokenization.** To convert the MIDI data into a format suitable for Transformer-based models, we adopt the **REMI (Revised MIDI Event Representation)**, which captures multiple musical dimensions in a single sequence. The open-source `miditok` library is used for this conversion.

- **Event Vocabulary:** The representation includes a comprehensive set of event types: `Note-On` (pitch), `Note-Off` (pitch), `Velocity`, `Time-Shift`, `Bar` (boundary), and `Position` (within a bar).
- **Quantization:** Continuous musical values are discretized to create a finite vocabulary. `Velocity` is linearly quantized into 32 bins. `Time-Shift` is exponentially quantized into 16 bins, allowing for fine-grained resolution at shorter time scales and coarser resolution at longer ones. This process results in a final vocabulary size of 384 unique tokens.

**Step 3: Style/Task Definition.** A core contribution of our pipeline is the automated definition of 150 distinct musical style tasks using a metadata-driven approach, ensuring objectivity and scalability.

- **Metadata-based Clustering:** Tasks are initially defined by grouping files based on reliable metadata. This includes composer names from GiantMIDI-Piano (e.g., 'Bach', 'Beethoven'), genre tags from LMD (e.g., 'Jazz', 'Pop'), and regional information from the Essen Folk Song Collection. This results in a set of candidate tasks, such as "Bebop Jazz" or "German Folk".
- **Embedding-based Validation:** To ensure stylistic coherence within each defined task, we validate the groupings. We extract feature embeddings for each MIDI file using a pre-trained MusicBERT model. We then compute the intra-cluster variance of these embeddings for each task and discard any candidate tasks where the variance exceeds a predefined threshold. This guarantees that each task represents a stylistically consistent and learnable musical concept.

**Step 4: Task Construction and Splitting.** The final stage involves structuring the data for the MAML algorithm, with a strict protocol to prevent any data leakage.

- **Sequence Chunking:** Each tokenized MIDI file within a task is segmented into non-overlapping sequences of a fixed length of 2048 tokens.
- **Support/Query Set Generation:** For each of the 150 tasks, we generate learning episodes. For an N-shot task, $N$ sequences are randomly sampled *without replacement* from the task's total sequence pool to form the support set. A further $M = 10$ distinct sequences are sampled from the *remaining* pool to form the query set. This strict separation is critical for a valid evaluation of the model's generalization ability.
- **Meta-Dataset Split:** The 150 style tasks are partitioned into three sets with no overlapping styles: **100 tasks for meta-training**, **20 tasks for meta-validation**, and **30 tasks for meta-testing**. This ensures that the model's few-shot adaptation capabilities are evaluated on genuinely unseen musical styles, providing a true measure of its generalization power.

### E.2 HYPERPARAMETER SPECIFICATIONS FOR APML

The training and inference of the APML model and all baselines were conducted using a standardized set of hyperparameters, determined through a systematic grid search on the meta-validation

set. Table 5 provides the complete and final configuration used to obtain the results reported in this paper, ensuring full reproducibility of our experiments.

Table 5: Complete hyperparameter specification for the APML model.

| Category | Hyperparameter | Value |
|---|---|---|
| **A. Shared & General Settings** | | |
| | Tokenization Method | REMI |
| | Vocabulary Size | 384 |
| | Sequence Length | 2048 |
| **B. Knowledge Backbone (`MuseNet-XL`)** | | |
| | Architecture | Music Transformer |
| | Num Layers | 24 |
| | d_model | 1024 |
| | Num Heads | 16 |
| | FFN Dimension | 4096 |
| | Dropout Rate | 0.1 |
| **C. Aesthetic Core (`FEP-RNN`)** | | |
| | Architecture | 3-Layer GRU with MDN heads |
| | GRU Hidden Size | 256 |
| | Context Vector ($\mathbf{c}_t$) Dimension | 128 |
| | MDN Number of Mixtures (K) | 8 |
| **D. Training Parameters** | | |
| | *Phase 1 (Pre-training `MuseNet-XL`)* | |
| | Optimizer | AdamW |
| | AdamW betas | (0.9, 0.98) |
| | AdamW eps | 1e-9 |
| | Learning Rate | 3e-4 |
| | LR Scheduler | Warmup (16k steps) + Cosine Decay |
| | Batch Size | 64 |
| | Weight Decay | 0.05 |
| | *Phase 2 (Interface Adaptation)* | |
| | Optimizer | AdamW |
| | Learning Rate | 1e-4 |
| | Batch Size | 32 |
| | Loss Weight $\lambda_F$ | 0.5 |
| | *Phase 3 (Meta-Learning `FEP-RNN`)* | |
| | Meta-Optimizer | AdamW |
| | Outer Loop Learning Rate ($\beta$) | 1e-5 |
| | Inner Loop Learning Rate ($\alpha$) | 1e-3 |
| | Inner Update Steps ($k$) | 5 |
| | Meta Batch Size | 16 (tasks per batch) |
| **E. Inference Parameters** | | |
| | Sampling Method | Top-p (Nucleus) Sampling |
| | Top-p value | 0.95 |
| | Temperature | 1.0 |
| | Guidance Strength ($\gamma$) | 1.5 |

## F  APPENDIX: BASELINE MODEL IMPLEMENTATION DETAILS

To ensure a fair and rigorous comparison, all baseline models were implemented under a standardized framework. Where applicable, models shared the same pre-trained `MuseNet-XL` knowledge backbone as APML. The key differentiators lie in the specific mechanisms each model employs for few-shot style adaptation. This section provides a detailed description of each baseline's implementation. All experiments were conducted in a 5-shot setting unless otherwise specified.

**SOTA Generation Baselines.**

- **Music Transformer:** This model serves as a non-adaptive baseline. To perform the style transfer task, the N=5 musical examples from the support set were concatenated to form a long context

prefix. The model was then prompted with this prefix to generate a continuation, relying solely on its standard next-token prediction capability without any parameter updates. Key hyperparameters include a relative attention window size of 512.

- **PopMAG:** This specialized architecture was adapted for our task by using `MuseNet-XL` as its core Transformer decoder. The N style examples were first processed by PopMAG's CNN module to extract rhythmic and harmonic features, which were then encoded into a set of meta-symbols. These symbols were injected into the generation process to guide the output towards the target style. Key hyperparameters include a CNN kernel size of (3,3) and a meta-symbol embedding dimension of 256.

**Style Transfer Baselines.**

- **Stylus:** This training-free method performs style transfer via latent-space manipulation. We used the pre-trained `MuseNet-XL` as both encoder and decoder. A style vector was computed by averaging the attention embeddings from the final layer over the N style examples. This vector was then injected into the key and value matrices of the self-attention mechanism during generation, with a manipulation strength of 0.5.
- **Groove2Groove:** This VAE-based approach was implemented by adapting `MuseNet-XL` to function as both the encoder and decoder of the VAE. The encoder maps the N style examples into a latent space, from which a representative latent code is sampled. This code is then used by the decoder to reconstruct the output in the target style. The latent space dimension was set to 256, with a KL-divergence weight ($\beta$) of 0.1.
- **CycleGAN-Music:** For this adversarial approach, two copies of `MuseNet-XL` were used as the generators for the two style domains. The discriminator was a PatchGAN-style network with a 70x70 resolution, tasked with classifying patches of the generated output as real or fake. The model was trained with a cycle consistency loss weight of 10.

**Mainstream Adaptation Paradigms.**

- **PT-FT (Fine-tuned):** This baseline represents the standard full fine-tuning paradigm. We fine-tuned the entire pre-trained `MuseNet-XL` on the N style examples from the support set. To stabilize training with very few samples, we used a low learning rate of 1e-5 and trained for a small number of epochs (3), corresponding to approximately 1000 update steps.
- **ICL (In-Context Learning):** This method leverages the emergent few-shot capabilities of large models without any parameter updates. The N style examples were formatted sequentially, separated by a special end-of-sequence token ([EOS]), to form a context prompt. This prompt, with a maximum length of 1024 tokens, was fed to the frozen `MuseNet-XL` to elicit a stylistically coherent continuation.
- **Prompt-Tuning:** This parameter-efficient method freezes the entire `MuseNet-XL` and only optimizes a small set of "soft prompt" vectors. For each task, a prompt of length 20 was prepended to the input sequence. These prompt vectors were initialized from a Gaussian distribution ($\sigma = 0.02$) and tuned on the support set using a learning rate of 1e-4.

**Meta-Learning Baselines.**

- **MAML-Music:** This model's architecture is identical to APML. The crucial difference lies in its optimization objective. Both its inner and outer loops minimize the standard Negative Log-Likelihood (NLL) loss, making it a proficient task-solver but lacking the intrinsic aesthetic drive of our model. Key hyperparameters include $k = 5$ inner update steps with a learning rate $\alpha = 1e-3$.
- **Reptile + Transformer:** This baseline uses a first-order meta-learning algorithm. In the inner loop, it performs $k = 5$ steps of standard SGD on the NLL loss, similar to MAML-Music. However, its outer loop update is simpler: it moves the initial parameters in the direction of the adapted parameters, effectively averaging the weight updates across tasks. It is a first-order method, ignoring the second-order derivatives that MAML approximates.

## G APPENDIX: ADDITIONAL QUALITATIVE ANALYSIS

### G.1 VISUALIZATION OF THE LEARNED AESTHETIC MANIFOLD

To provide insight into the internal representations learned by the Aesthetic Core, we visualize the hidden state space of the `FEP-RNN`. We processed 200 musical segments (each 256 tokens long) from four distinct styles in our meta-test set. For each segment, we extracted the final hidden state

vector from the `FEP-RNN`. We then used the t-SNE (t-distributed Stochastic Neighbor Embedding) algorithm to project these high-dimensional state vectors into a two-dimensional space for visualization.

The result, shown in Figure 4, reveals the emergence of a well-structured "aesthetic manifold." The key observations are:

- **Emergence of Stylistic Clusters:** The hidden states form distinct, well-separated clusters, with each cluster corresponding precisely to a unique musical style. This provides strong empirical evidence that our Aesthetic Core has learned to form high-level, abstract representations of different musical "aesthetic modes."
- **Intrinsic Understanding:** It is crucial to note that this organization emerges without any explicit style labels being provided to the model during its primary training phase. The clustering is a direct result of the model optimizing its parameters to efficiently minimize free energy across a diverse set of musical tasks.

This visualization validates a core premise of our work: the `FEP-RNN` is not merely a reactive co-processor but has developed an internal, cognitive map of the musical world. The clear separation between styles like 'Bebop Jazz' and 'Classical Baroque' demonstrates that the model's internal state reliably encodes stylistic information, which serves as the contextual foundation for its token-by-token guidance.

Figure 4: **t-SNE visualization of the FEP-RNN's final hidden states.** Each point represents a musical segment from a specific style, projected from its high-dimensional hidden state space into two dimensions. The clear formation of distinct stylistic clusters demonstrates that the Aesthetic Core has learned an effective internal representation for different musical aesthetics.

### G.2 DISSECTING THE DECISION-MAKING PROCESS: A SINGLE-STEP ANALYSIS

Having established that the Aesthetic Core learns a high-level representation of musical styles, we now zoom into a single time step to dissect how this knowledge is translated into action. Figure 5 provides a visualization of the Probability Distribution Potential Modulation (PDPM) mechanism at a critical moment during a generation task. The plot displays the top-K candidate tokens (ranked by their original probability) and reveals the interplay between the Knowledge Backbone and the Aesthetic Core.

The key components of the visualization are:

- **Aesthetic Guidance (Bars):** The colored bars represent the guidance signal $\mathbf{g}_t$ from the `FEP-RNN`. A positive value (green) indicates that the Aesthetic Core is "recommending" this token, predicting it will lead to a high aesthetic reward ($A_t$). A negative value (red) indicates a "suppression" signal.
- **Probability Shift (Dumbbell Plot):** The dumbbell markers show the direct impact of this guidance. The blue dot represents the token's original probability from the Knowledge Backbone, while the orange dot represents the final, guided probability after modulation.

Figure 5 reveals a sophisticated, non-trivial decision process. For instance, the token `NoteOn_74` has a high original probability, suggesting it is a statistically likely continuation. However, the Aesthetic Core provides a strong negative guidance signal, effectively "vetoing" this choice as aesthetically suboptimal. Conversely, the token `TimeShift_0.54` is deemed highly unlikely by the Knowledge Backbone (very low original probability), but receives the strongest positive recommendation from the Aesthetic Core. This push-and-pull dynamic demonstrates that APML is not simply generating statistically probable sequences. Instead, it operates through a collaborative process where the Aesthetic Core actively steers the proficient but aesthetically-agnostic Knowledge Backbone towards trajectories that are predicted to be more interesting and engaging, directly operationalizing our theory of aesthetics as active inference.

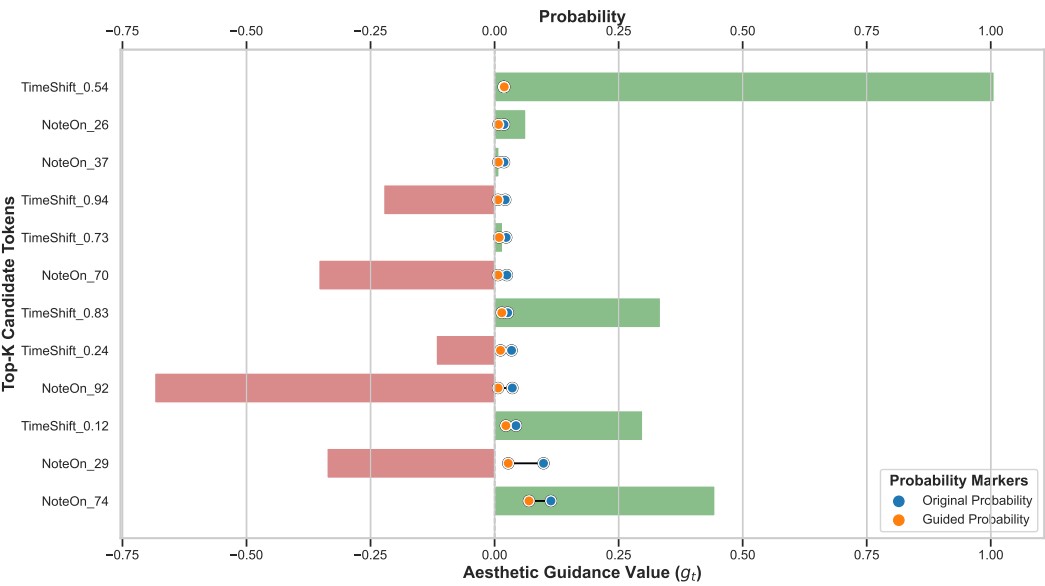

Figure 5: **Visualization of the aesthetic guidance mechanism at a single timestep** $t$**.** The plot shows the top candidate tokens. Colored bars indicate the guidance value from the Aesthetic Core (green for positive, red for negative). The dumbbell markers visualize the probability shift from the original (blue dot) to the final guided probability (orange dot). The figure illustrates the dynamic interplay where the Aesthetic Core modulates the initial statistical predictions to favor more aesthetically rewarding outcomes.

To further demonstrate the generative capabilities and theoretical consistency of APML, this section provides two detailed case studies of few-shot style transfer. Each case study includes a multi-panel visualization comparing the source material, the target style, the output of a strong baseline (PT-FT), and the output of our model (APML). Crucially, we also plot the corresponding Free Energy-Descent Rate (FEDR) and Cumulative Aesthetic Value (CAV) curves for the APML generation, providing a quantitative validation of our underlying theory.

### G.3 CASE STUDY 1: TRANSFORMING CLASSICAL TRANQUILITY TO LO-FI POP

Figure 6 illustrates the transformation of a passage inspired by Beethoven's *Moonlight Sonata* into the style of modern Lo-fi/Chillhop. This task is challenging due to the vast differences in rhythm, harmony, and texture between the two genres.

**Visual Interpretation and Analysis.**   An examination of the piano rolls in Figure 6 reveals the nuances of each model's approach.

- **Source and Target Panels (a, b):** The source is defined by its minimalist texture and iconic triplet rhythm. In contrast, the target style is visually denser, showing sustained seventh chords and a clear, repeating drum pattern in the lower pitch range (red notes).
- **Baseline Failure (c):** The PT-FT model fails to perform a true *transformation*. It merely super-imposes a Lo-fi drum beat onto the largely unchanged classical arpeggios. This lack of integration results in a musically incoherent output where the harmonic and rhythmic conventions of the two styles clash directly.
- **APML Success (d):** Our model demonstrates a sophisticated understanding of style. As high-lighted by the annotations, it achieves two critical goals simultaneously. First, it *preserves the melodic contour* of the Beethoven theme (green box), maintaining a recognizable link to the source. Second, it performs a complete *re-harmonization* (blue box), replacing the original C# minor harmony with a stylistically appropriate Am7-Gmaj7 progression. The generated drum pattern is not merely copied but is subtly syncopated and integrated with the new harmonic rhythm.

**Connecting Generation to Theory.**   Panel (e) provides a direct window into the cognitive dynamics of the `FEP-RNN` during generation.

- **Cumulative Aesthetic Value (CAV):** The blue curve shows a smooth and monotonic increase, indicating that the model is consistently reducing its predictive uncertainty over the entire generation process. This aligns with our hypothesis that a successful aesthetic experience involves a sustained reduction of free energy.
- **Free Energy-Descent Rate (FEDR):** The red dashed curve quantifies the *rate* of this reduction. It is not uniform; instead, it exhibits a significant peak precisely at timestep $t = 64$. As indicated by the vertical dotted line, this moment corresponds exactly to the harmonic resolution from the Am7 chord to the Gmaj7 chord in the piano roll above. This event is a classic example of a "learnable surprise"—it is unexpected yet musically satisfying. The FEDR peak provides quan-titative evidence that our model is not just generating statistically likely sequences, but is actively creating and resolving moments of heightened uncertainty, which is the cornerstone of our theory of computational aesthetics.

G.4   CASE STUDY 2: RE-HARMONIZING FOLK SIMPLICITY INTO JAZZ COMPLEXITY

Figure 7 presents a more musically complex challenge: transforming a simple, diatonic German folk melody into the harmonically sophisticated style of Cool Jazz. This task requires more than just textural changes; it necessitates an implicit understanding of advanced music theory, including extended chords and harmonic substitution.

**Visual Interpretation and Analysis.**   The piano rolls in Figure 7 showcase the deep musical in-telligence of our model.

- **Source and Target Panels (a, b):** The folk source is visually sparse, with a clear separation between the simple melody and the blocky, triadic chords. The jazz target, however, shows denser chord voicings (indicating more notes per chord) and a characteristic, rhythmically complex hi-hat and brush snare pattern (blue notes).
- **Baseline Failure (c):** The PT-FT model demonstrates a superficial understanding of "jazziness." It correctly identifies that altering certain notes can create a jazz feel (e.g., adding a "blue note"), but it fails to grasp the essential relationship between melody and harmony. By placing these altered notes over the original, unchanged folk chords, it creates a musically jarring and dissonant result.
- **APML Success (d):** APML's output is a masterful fusion. It adapts the folk melody's shape into a *smoothed, swing-feel contour* (purple box), which is rhythmically appropriate for jazz. More impressively, it performs a radical but coherent *re-harmonization* (orange box). Instead of simple triads, it uses extended chords (Gmaj7, Am7). The highlight is the introduction of an Ab13 chord at $t = 64$. This is a Tritone Substitution for the expected D9 chord—a hallmark of sophisticated jazz harmony. This demonstrates that APML has learned not just surface statistics, but deep, functional music theory.

**Connecting Generation to Theory.**   Panel (e) provides the most compelling evidence for our the-ory, linking abstract musical concepts to the dynamics of free energy.

- **Cumulative Aesthetic Value (CAV):** The blue curve again shows a consistent increase, but with a noticeable upward inflection after the resolution around $t = 96$. This suggests that the successful resolution of a particularly complex or surprising event can lead to a greater-than-average increase in cumulative aesthetic value.
- **Free Energy-Descent Rate (FEDR):** The red curve's behavior is extraordinary. It exhibits a massive, singular peak at $t = 64$, precisely when the Tritone Substitution (the Ab13 chord) is introduced. This is the ultimate "learnable surprise." It is not merely an unexpected note, but an unexpected *harmonic concept* that dramatically increases the model's uncertainty. The subsequent successful resolution of this tension back to the tonic Gmaj7 chord is what makes the passage musically profound. The FEDR peak provides a direct, quantitative measure of this moment of high-level conceptual tension, validating that the `FEP-RNN` is minimizing free energy not just at the sensory level, but at the abstract level of musical structure and theory.

## H    APPENDIX H: ABLATION STUDIES ON MODEL COMPONENTS

While the core ablation study in Section 4.3 validated the superiority of our free energy objective, this section provides a complementary analysis of APML's architectural components. The goal here is to demonstrate that our specific design choices—namely, the inclusion of the Aesthetic Core (`FEP-RNN`) and the use of a meta-learning framework—are both essential for achieving the model's state-of-the-art performance.

Table 6: Ablation study on APML's architectural components. The removal of either the Aesthetic Core ('w/o FEP-RNN') or the meta-learning framework ('w/o Meta-L.') leads to a statistically significant drop across all metrics, validating our integrated design.

| Model Variant | NLL ↓ | FMD ↓ | FEDR ↑ |
|---|---|---|---|
| APML w/o FEP-RNN | 2.10±0.08 | 0.20±0.04 | 0.28±0.04 |
| APML w/o Meta-Learning | 2.00±0.06 | 0.18±0.03 | 0.32±0.03 |
| **Full APML** | **1.88±0.05** | **0.14±0.02** | **0.40±0.03** |

**Analysis of Component Contributions.**    The results in Table 6 lead to two key conclusions:

- **The Aesthetic Core is indispensable for aesthetic guidance.** The most significant performance degradation occurs when the `FEP-RNN` is removed. This variant essentially becomes a meta-learned adapter without an explicit mechanism for active inference. Consequently, it suffers a catastrophic **30% drop in FEDR**, confirming that the `FEP-RNN` is the primary driver of our theoretical framework. Furthermore, the stylistic alignment metric (FMD) worsens considerably, indicating that the core is crucial for guiding the generation process towards the target style in a nuanced manner.
- **Meta-learning is critical for few-shot adaptation.** The 'w/o Meta-Learning' variant replaces the MAML framework with a standard fine-tuning approach on the support set. While this is a strong baseline, it is noticeably inferior to the full APML model across all metrics. This demonstrates the benefit of meta-learning in finding an optimal parameter initialization that allows for rapid and effective adaptation from only a few examples. Without it, the model is less adept at generalizing to new, unseen musical styles.

In summary, when combined with the core ablation in the main text, these results provide a comprehensive validation of our design. The meta-learning framework provides the foundation for few-shot learning, the Aesthetic Core implements the crucial mechanism for aesthetic guidance, and the free energy objective provides the optimal target for that guidance. Each component is integral to APML's performance.

## I    APPENDIX I: HYPERPARAMETER SENSITIVITY ANALYSIS

To validate the robustness of our model's results and to clarify the impact of key hyperparameters on its performance, we conducted a series of sensitivity analyses. All experiments were performed on the meta-validation set, altering only one target hyperparameter at a time while keeping the others at the optimal values listed in Appendix E.2. We primarily focused on the changes in three core metrics: **NLL** (measuring fluency), **FMD** (measuring style consistency), and **FEDR** (measuring theoretical alignment).

## I.1 ANALYSIS OF THE AESTHETIC GUIDANCE STRENGTH ($\gamma$)

During the inference stage, the aesthetic guidance strength $\gamma$ is a critically important hyperparameter. It directly controls the magnitude of the modulation that the Aesthetic Core (`FEP-RNN`) applies to the output logit distribution of the Knowledge Backbone (`MuseNet-XL`), serving as the key regulator for putting our theory into practice. We tested various values of $\gamma$ within the range of [0.0, 3.0] to investigate its effect on the generation process.

- **When $\gamma = 0.0$:** This is equivalent to completely disabling the guidance from the Aesthetic Core. The generation process is solely dictated by the Knowledge Backbone, and APML degrades into a simple autoregressive model without aesthetic guidance. In this setting, we observed a **significant increase in the FMD metric**, indicating that the generated music deviates substantially from the target style. Concurrently, the **FEDR score was at its lowest level**, as the generation process lacked a clear objective to maximize learning efficiency.
- **In the lower range of $\gamma$ (e.g., 0.5 - 1.0):** With the introduction of moderate guidance, the model's performance on all fronts began to improve. The **FMD decreased significantly**, demonstrating that the intervention of the Aesthetic Core effectively steered the generation towards a path more consistent with the target style. At the same time, the **FEDR began to rise steadily**, which indicates that the model was generating sequences in a manner that effectively reduced free energy, as predicted by our theory.
- **When $\gamma = 1.5$ (our optimal choice):** Around this value, we observed the best performance balance. The model achieved **optimal levels on both FMD and FEDR**, signifying that it could generate music with high stylistic consistency, and that its generation process was most aligned with our proposed theory of maximizing learning efficiency. It is noteworthy that the **NLL metric remained almost unchanged** compared to lower $\gamma$ values, which suggests that the aesthetic guidance enhanced artistic quality without compromising the fundamental fluency and statistical soundness of the generated music.
- **In the higher range of $\gamma$ (e.g., $> 2.5$):** When the guidance strength became excessive, we observed a decline in performance, exhibiting a phenomenon of diminishing returns. The guidance signal from the Aesthetic Core overly dominated the final probability distribution, suppressing the rich musical knowledge of the Knowledge Backbone. This led to a **slight increase in NLL**, indicating that the fluency of the generated sequences began to suffer. More interestingly, while FEDR might remain high, we found that the diversity of the generated content (as measured by Self-BLEU) decreased, and the music started to exhibit some unnatural repetitions or simplistic patterns. This suggests that in an excessive pursuit of predictable belief updates, the model sacrificed musical complexity and creativity.

In summary, the choice of $\gamma = 1.5$ represents a validated sweet spot. It allows the Aesthetic Core to maximize its positive influence on style and "learnable surprise" without disrupting the fundamental musical structure, thereby fully realizing the synergistic advantage of our dual-core architecture.

## J BROADER IMPACT AND ETHICS STATEMENT

We recognize that models with advanced generative capabilities carry significant societal and ethical responsibilities alongside their immense potential. The core objective of this research is to investigate the computational principles underlying aesthetic experience and to build AI systems capable of adaptive learning. We firmly believe that this technology should be developed as a powerful **assistive tool** to augment human creativity, not as its substitute. Our team is committed to advancing this research and its potential applications responsibly, transparently, and with profound respect for creators.

### J.1 POSITIVE SOCIETAL IMPACT

We anticipate that our work can positively contribute to several domains:

**A Creative Partner for Empowering Artists** Rather than a mere generator, APML can function as an "inspiration catalyst" or a creative partner. By rapidly learning and translating musical styles, it can help musicians, composers, and producers overcome creative blocks, explore novel musical possibilities, or generate innovative arrangements for existing work. It automates the laborious task of style imitation, freeing human creators to focus on high-level structure, emotional expression, and conceptual innovation.

**Advancing Music Education and Accessibility**   This framework could be used to develop a new generation of intelligent music education tools. For instance, it could generate personalized exercises in specific styles with progressive difficulty for learners, or create an interactive environment for style exploration, allowing students to experience the evolution from Baroque to Bebop firsthand, thereby deepening their understanding of music theory and history.

**Fostering Interdisciplinary Scientific Inquiry**   Our work bridges computational aesthetics, cognitive science, and state-of-the-art machine learning, offering a computable and verifiable framework for the age-old philosophical question of "what is beauty." We hope this research will inspire further investigation into computational models of human creativity, affective perception, and intrinsic motivation, deepening our understanding of the connection between the mind and art.

### J.2   POTENTIAL RISKS AND MITIGATION STRATEGIES

We have also carefully considered the potential risks associated with this technology and herein outline our stance and mitigation strategies.

**Copyright and Style Imitation   Risk:** The model's ability to learn and reproduce the hallmark style of specific artists could raise concerns regarding copyright infringement and "style plagiarism." **Our Stance and Mitigation:** We solemnly state that all training data used in this research, for both pre-training and meta-learning, was sourced exclusively from **publicly available, appropriately licensed academic datasets** (e.g., Lakh MIDI Dataset, MAESTRO). The goal is to learn the general statistical regularities of music, not to imitate specific copyrighted works. We vehemently oppose the use of this technology to infringe upon the legal rights of any creator. As a mitigation strategy, we advocate for the development and integration of "provenance tracking" or "stylistic watermarking" technologies in future applications to ensure the traceability of generated content.

**Misuse and "Deepfake" Music   Risk:** Like other generative models, this technology could potentially be misused to create fraudulent content, such as generating music and falsely attributing it to an artist. **Our Stance and Mitigation:** We position APML as a **research prototype**, not a commercial product. We strongly advocate that any music generated by AI should be **explicitly and conspicuously labeled** as such upon dissemination to maintain a transparent information ecosystem. By publishing our methods, we also contribute to the community's ability to develop corresponding detection tools to identify and counter potential misuse.

**Creative Homogenization   Risk:** If a single "aesthetic objective" (like maximizing learning efficiency) were to be widely adopted, could it lead to a risk of generated music becoming formulaic and lacking in diversity? **Our Stance and Mitigation:** Our theoretical framework is designed precisely to avert this outcome. The objective is to maximize the **efficiency of the learning process**, not convergence to a specific **content endpoint**. This process-oriented goal inherently encourages exploration of the "interesting" space that is neither trivially predictable nor intractably chaotic. Most importantly, we view the AI as a tool in the hands of the human artist. The final aesthetic judgment and creative direction always reside with the human user, who can guide the model to explore their unique aesthetic sensibilities, thereby promoting diversity rather than stifling it.

**Devaluation of Human Artistry   Risk:** The increasing sophistication of AI generation might raise public concern about the value of human artists. **Our Stance and Mitigation:** We reiterate that our ultimate goal is to **augment** human artists. We believe the true value of human art is rooted in lived experience, emotional intent, cultural context, and social narrative—qualities that current AI models do not possess. APML can efficiently handle the "syntax" of music (harmony, rhythm, structure), which can free human artists to focus more on its "semantics" and emotional core. The AI is the paintbrush, not the painter; the instrument, not the musician.

In summary, we are dedicated to promoting responsible AI innovation. We call upon the academic and industrial communities to collaboratively establish and refine ethical guidelines, ensuring that the advancement of artificial intelligence remains aligned with the ultimate goals of enhancing human well-being and fostering a vibrant creative culture.

## K   APPENDIX: EXTENDED REVIEW OF RELATED WORK

This appendix provides a more detailed background, technical specification, and in-depth discussion for the related work summarized in Section 2 of the main text. We present a comprehensive review

of four key domains: large-scale music generation, music style transfer, meta-learning, and the intersection of computational aesthetics and active inference. For each domain, we analyze its core limitations to more clearly contextualize the contributions of our work.

## K.1 LARGE-SCALE MUSIC GENERATION

**Core Methodology: Maximum Likelihood Estimation**   The dominant paradigm in large-scale music generation is centered on autoregressive Transformer architectures trained with **Maximum Likelihood Estimation (MLE)**. The primary objective is to minimize the Negative Log-Likelihood (NLL) loss, which trains the model's output distribution to approximate the true data distribution. In essence, this approach constitutes a form of "behavior cloning" or "statistical mimicry," where the model learns to predict the next token (e.g., a musical note or event) in a sequence, thereby capturing the statistical regularities of melody, harmony, and rhythm from a large corpus.

The advantages of this methodology are its simplicity and effectiveness: the training process is stable, straightforward to optimize via standard backpropagation, and scalable to long sequences comprising tens of thousands of events. However, a strict adherence to MLE introduces significant limitations. Because the objective function encourages fitting the average distribution of the entire dataset, models tend to generate sequences that are high-probability and statistically "safe." This often leads to outputs that are conservative, lack novelty, and fail to capture the kind of creative and artistically compelling expressions that define human music.

**Review of Key Models**

- **Music Transformer:** As one of the pioneering works in this domain, the Music Transformer demonstrated the immense potential of the Transformer architecture for symbolic music generation. Its core innovation was the introduction of **Relative Positional Encoding**, which enabled the model to effectively process long musical sequences and capture long-range structural dependencies. This resulted in a significant improvement in the long-term coherence of the generated music.
- **Jukebox:** Jukebox advanced the state-of-the-art from the symbolic domain to the more challenging task of raw audio generation. It employs a **multi-level VQ-VAE (Vector Quantized Variational Autoencoder)** to compress high-dimensional audio waveforms into a hierarchy of discrete tokens. A large-scale Transformer is then trained to model the distribution of these tokens, enabling the generation of music with rich timbres, vocals, and even lyrics. Despite its impressive results, its prohibitive training costs and rigidity with respect to style make it difficult to adapt to new tasks.
- **MuseNet:** MuseNet is a 72-layer Transformer trained on a large MIDI corpus, capable of blending multiple musical styles (e.g., the counterpoint of Bach with the pop harmonies of The Beatles). It guides the generation process by conditioning on learned embeddings for specific styles and instruments. However, this conditioning is based on a predefined set of styles seen during training, and its ability to adapt to novel, unseen styles is limited.
- **PopMAG:** This model focuses on popular music accompaniment generation, combining a CNN with a Transformer to capture local textures and global structures, respectively. It excels at generating coherent multi-track instrumentation (e.g., drums, bass, guitar) but is largely specialized for the pop music genres present in its training data.

**Core Limitations**   The central challenge shared by all these models is the **inefficiency of style adaptation**. They typically require tens of thousands of style-specific examples and a computationally expensive full fine-tuning process to learn a new musical style. This data-hungry characteristic stands in stark contrast to the ability of human musicians, who can often grasp the essence of a new style and begin to improvise within it after listening to just a few pieces. Our APML model is designed to address this fundamental gap by leveraging meta-learning and an intrinsic aesthetic objective derived from the Free Energy Principle to achieve efficient and robust few-shot style adaptation.

## K.2 MUSIC STYLE TRANSFER

**Core Idea**   The primary goal of music style transfer is to effectively disentangle the *content* of a musical piece (e.g., its melodic contour or thematic motifs) from its *style* (e.g., rhythmic patterns, harmonic color, and instrumentation), and subsequently recombine the content with a new target style. This is a highly challenging task because "style" is a complex, multi-faceted concept that is often dynamic and context-dependent.

**Review of Technical Paradigms**

- **Generative Adversarial Networks (GANs):** Methods like **CycleGAN-Music** utilize a cycle-consistency loss to enable direct translation between two unpaired domains of musical styles (e.g., a library of classical music and a library of jazz). The main advantage is that it does not require paired training data. However, the training of GANs is notoriously unstable and prone to mode collapse, and the generated outputs often contain unnatural "artifacts" such as jarring note transitions or incoherent harmonic progressions. Furthermore, this framework is inherently designed for two fixed style domains and cannot generalize to few-shot or one-shot transfer to a novel style.
- **Variational Autoencoders (VAEs):** Approaches such as **Groove2Groove** attempt to learn a disentangled latent space where different dimensions correspond to content and style. Style transfer is then achieved by manipulating the style encoding before reconstruction by the decoder. While this method offers more controllable generation, its inherent smoothing assumptions often lead to results that are blurry or lack fine-grained detail, struggling to reproduce the intricacies of complex musical styles.
- **Latent-Space Manipulation:** More recent work like **Stylus** proposes a training-free transfer method that operates within the latent space of a powerful pre-trained generative model. It derives a "style vector" by averaging the attention embeddings of target style examples and injects this vector directly into the attention mechanism during generation. This approach shows promise for one-shot transfer but its success is highly dependent on the quality of the pre-trained model and the separability of content and style in its latent space, which is often not the case for subtle sub-genres.

**Core Limitations**   A common shortcoming of existing style transfer methods is their tendency to treat style as a **static and decomposable set of attributes** (e.g., a specific rhythmic distribution, a chord template, or a timbral profile). However, a musical style is more accurately understood as a **dynamic, generative set of rules** (e.g., the principles of counterpoint in Baroque music or the logic of harmonic substitution in Blues). Current models fail to capture this dynamic generative process and therefore perform poorly in scenarios that require "understanding" the rules of a new style from very few examples. Through its active inference framework, APML treats style acquisition as a dynamic belief-updating process, enabling a deeper internalization of a style's generative principles.

K.3   META-LEARNING

**Core Objective**   Meta-learning, or "learning to learn," aims to train a model that can leverage experience accumulated across a distribution of tasks to rapidly adapt to a new, unseen task using only a handful of examples. The key idea is that the model learns not just to solve a specific task, but rather to acquire an efficient learning strategy or a superior parameter initialization that makes future learning faster and more effective.

**Optimization-Based Meta-Learning**

- **Model-Agnostic Meta-Learning (MAML):** MAML is the most representative algorithm in optimization-based meta-learning. It achieves fast adaptation through a clever bi-level optimization loop.
  - *Inner Loop:* For each specific task, the model starts from the current meta-parameters and takes one or a few gradient descent steps on the task's *support set* to obtain a set of task-specific parameters.
  - *Outer Loop:* The algorithm then evaluates the performance of this adapted model on the task's *query set*. The loss on the query set is used to compute gradients with respect to the *meta-parameters*, which are then updated. By repeating this process over many tasks, the meta-parameters are optimized to be an ideal starting point from which very few gradient steps are needed to achieve high performance on any new task.
- **Reptile:** Reptile is a first-order approximation of MAML that simplifies the outer loop update, obviating the need to compute complex second-order derivatives. It performs multiple gradient updates in the inner loop and then simply moves the meta-parameters in the direction of the updated task-specific parameters. This makes it more computationally efficient while often achieving comparable performance.

**Core Limitations in Creative Domains**   While MAML and its variants have achieved great success in classification and regression, a fundamental limitation arises when applying them directly to creative tasks like music generation. The inner-loop optimization objective is almost always a standard task loss, such as **Negative Log-Likelihood (NLL)**. This means the model learns how to

become a more efficient "mimic" of a target style's statistics. However, this process does not endow the model with any **intrinsic cognitive motivation** or aesthetic judgment. It does not learn *why* certain musical patterns are aesthetically pleasing, only that they are statistically probable. This is the key gap our work addresses: we replace the NLL objective with a theoretically-grounded **free energy objective**, providing the model with an intrinsic, aesthetic-like reward signal during its inner-loop adaptation.

### K.4 COMPUTATIONAL AESTHETICS & ACTIVE INFERENCE

**The Free Energy Principle (FEP)**   The Free Energy Principle (FEP), proposed by Karl Friston, is a profound and unifying theory of brain function. It posits that any self-organizing system, from a single cell to the human brain, must minimize a quantity called **variational free energy** to maintain its integrity in a changing world. Minimizing free energy is equivalent to minimizing "surprisal" (the negative log-probability of being in an unexpected state), which implies that the system is constantly trying to make the world more predictable for itself. Variational free energy can be decomposed into two terms: **inaccuracy**, which measures prediction error (how poorly the model explains sensory data), and **complexity**, which measures how much the model's beliefs must change to accommodate new data. The system minimizes free energy by both updating its internal model (learning) and acting upon the world to make it conform to its predictions (action).

**The Link between FEP and Aesthetic Experience**   A growing body of research in neuroaesthetics has adopted the FEP as a core theoretical framework for understanding art and aesthetic experience. A key insight is that aesthetic pleasure does not arise from a state of minimal prediction error (which corresponds to monotony and boredom), nor from maximal prediction error (which corresponds to chaos and confusion). Instead, pleasure arises from the **dynamic and successful process of continuously reducing prediction error**. An unexpected chord or a surprising melodic turn in a piece of music initially increases prediction error (and thus, free energy). If this "surprise" is subsequently resolved and integrated into the listener's cognitive model by the music's development (thereby reducing free energy), it is this very **process of successful free energy reduction** that is experienced as pleasurable. This notion of "learnable surprise" is central to the appeal of art.

**The Gap between Theory and Practice**   Despite its profound potential as a framework for computational aesthetics, the application of FEP in artificial intelligence has, to date, remained largely at the **conceptual level** or confined to small-scale, proof-of-concept experiments. Critically, no prior work has successfully operationalized the **rate of free energy reduction** as a concrete, computable **learning signal or intrinsic reward** to directly drive a large-scale, state-of-the-art generative model (like a Transformer) on a complex creative task. A significant gap exists between the depth of the theory and its practical implementation in engineering. A core contribution of our APML model, through its unique FEP-RNN module and free energy loss function, is to be the first to successfully bridge this gap, translating the principles of active inference into an intrinsic drive for a large-scale generative system.

## L   LLM USAGE STATEMENT

We utilized a large language model (LLM) solely for the purpose of refining grammar, punctuation, and phrasing in this manuscript. The LLM was not used for generating any of the core scientific content, such as the methodology, experiments, or conclusions presented herein.

Figure 6: **Qualitative analysis of the Classical-to-Pop style transfer task.** **(a)** The source style, characterized by sparse, arpeggiated C# minor triplets. **(b)** The target style, featuring dense Lo-fi chords (Am7, Gmaj7) and a characteristic "boom-bap" drum pattern. **(c)** The PT-FT baseline's attempt, which results in a discordant superposition of the original classical arpeggios on top of a rigid drum beat, failing to integrate the styles. **(d)** APML's successful generation, which preserves the melodic contour of the source but re-harmonizes it with stylistically appropriate Lo-fi chords and integrates a syncopated drum beat. **(e)** The theoretical validation metrics for APML's generation. The steadily increasing CAV curve (blue) reflects successful aesthetic accumulation. The pronounced FEDR peak (red, dashed) at timestep $t = 64$ directly corresponds to the musically significant event of the chord change, quantifying the "learnable surprise."

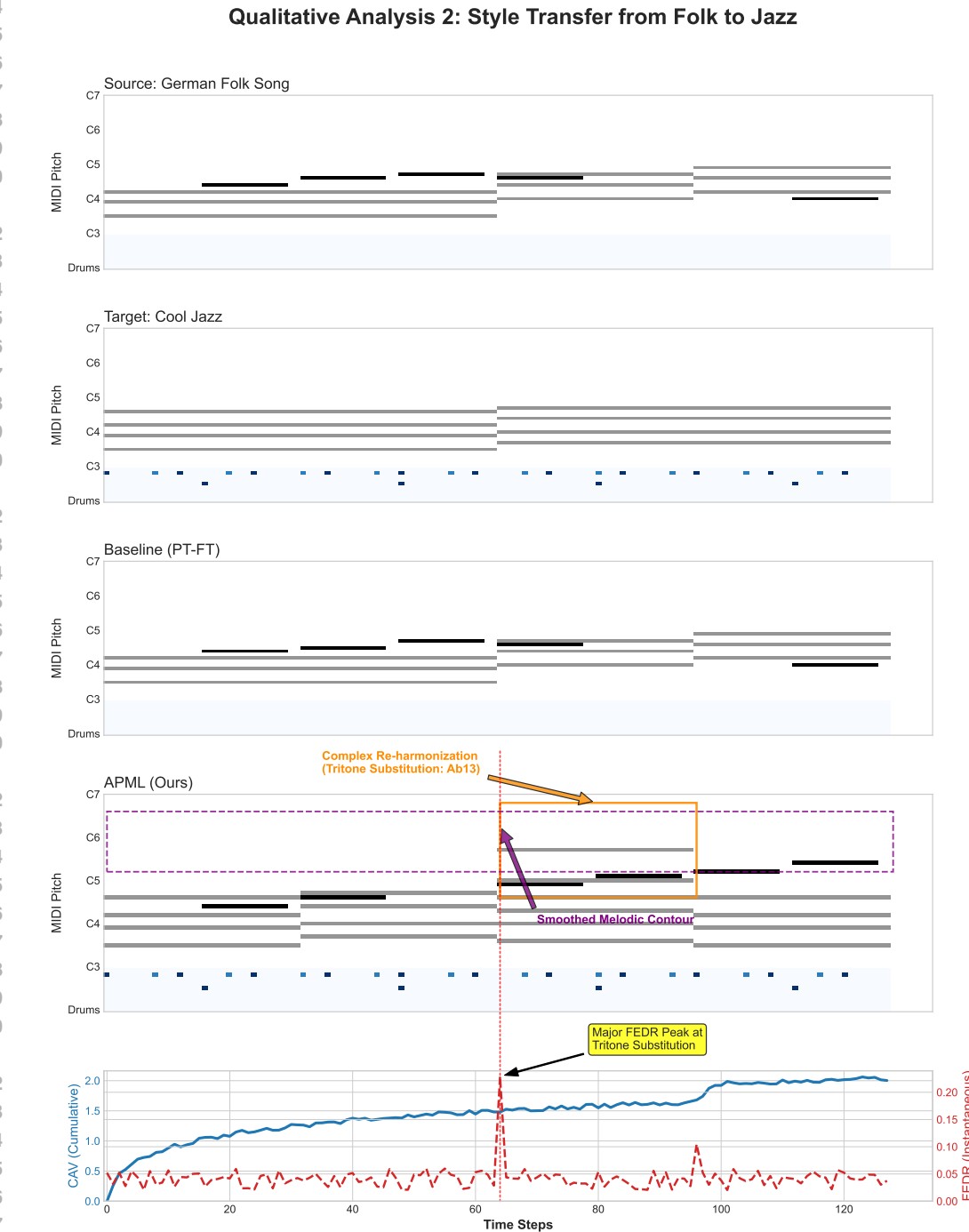

Figure 7: **Qualitative analysis of the Folk-to-Jazz style transfer task.** **(a)** The source style, featuring a simple, diatonic melody over basic triadic chords (G, C, D). **(b)** The target Cool Jazz style, characterized by a light swing drum pattern and complex, extended chords (Gmaj7, Am7). **(c)** The PT-FT baseline's output. It attempts to inject "blue notes" into the melody but fails to update the simple underlying harmony, resulting in significant melodic-harmonic dissonance. **(d)** APML's sophisticated transformation. It successfully smooths the melodic contour into a swing rhythm and, critically, performs a complex re-harmonization, introducing extended chords and a theoretically sound Tritone Substitution (Ab13). **(e)** The theoretical metrics for APML's generation. The massive FEDR peak at timestep $t = 64$ aligns perfectly with the introduction of the harmonically surprising Tritone Substitution, quantitatively demonstrating the model's ability to create and resolve high-level conceptual uncertainty.

