# OpenReview forum: "Why Music Moves Us: A Computational Model of Aesthetic Experience and Creativity via Meta-Learned Active Inference"
_ICLR.cc/2026/Conference — Submitted to ICLR 2026_

### Official Review · Reviewer_f2bm · 2025-10-31

**Soundness:** 3
**Presentation:** 2
**Contribution:** 3
**Rating:** 4
**Confidence:** 3

**Summary:**

This paper introduces the Aesthetic Priors Meta-Learner (APML), a generative framework designed to make aesthetic judgments grounded in musical knowledge. The authors clearly motivate the problem and situate their work in existing literature. Section 3 explains key theoretical foundations, including the Free Energy Principle (in terms of inaccuracy and complexity), aesthetic pleasure, and meta-learning. Section 4 then presents APML's system architecture, free-energy proxy implementation, and training approach. For evaluation, the authors construct a benchmark based on publicly available music data. Experimental results indicate that APML outperforms baseline methods across most metrics.
The main contribution of this work is its computational formulation of aesthetics through APML, applying the Free Energy Principle to conceptualize aesthetic pleasure as the rate of error reduction. The paper successfully operationalizes a complex theoretical framework into a coherent and implementable model.

**Strengths:**

•	The work demonstrates strong originality by applying the Free Energy Principle to aesthetics, framing aesthetic pleasure as error reduction over time.
•	The theoretical framework is effectively translated into a concrete architecture with a clear, principled training objective.
•	The writing is generally clear, with well-defined terminology and logically structured exposition that aids understanding of the technical components.
•	APML effectively models learnable musical structures, such as harmonic resolution and rhythmic predictability, suggesting a strong link between computational surprise and musical form.

**Weaknesses:**

-The paper adopts a relatively narrow definition of aesthetics. Although the evaluation employs diverse technical metrics (FMD, TC, CPS, FEDR, CAV), these primarily capture structure, learnability, and stylistic coherence. They do not reflect subjective, emotional, or cultural dimensions of aesthetic experience.
-The evaluation primarily validates the computational hypothesis, not the psychological claim that aesthetic pleasure corresponds to error-reduction. Without user studies, the results support APML as a model of stylistic learning efficiency, rather than a comprehensive model of aesthetics.
-While large-scale user studies may be planned, a small-scale preliminary human evaluation would substantially strengthen the paper. See, for example: “An Empirical Study on How People Perceive AI-Generated Music,” CIKM 2022.

**Questions:**

•	Minor presentation issue: In Figure 1, the lines for Free Energy and Aesthetic Pleasure overlap with full opacity, making them difficult to distinguish. Adjusting color contrast or styling would improve interpretability of this core conceptual illustration.
•	The introduction does not provide a clear overview or summary of the work, making it less engaging and leaving readers uncertain about the paper’s main focus.
•	LLMs could potentially serve as a more powerful backbone model and may improve performance. Including experiments with LLM-based backbones would help strengthen the empirical evaluation.
•	If the model optimizes for maximal error reduction, there may be a risk of reduced aesthetic diversity, biasing generation towards predictable and formulaic music, rather than creativity or novelty.

---

### Official Review · Reviewer_bhj6 · 2025-10-31

**Soundness:** 4
**Presentation:** 4
**Contribution:** 3
**Rating:** 8
**Confidence:** 5

**Summary:**

This paper addresses the fundamental aesthetic question of "why music is beautiful" by proposing a computable theoretical framework grounded in the Free Energy Principle (FEP) and Active Inference. It defines aesthetic pleasure as the rate of variational free energy reduction (explaining why monotonous sounds (no free energy descent) and chaotic noise (no systematic descent) are unaesthetic, while engaging music sustains this descent. To operationalize the framework, the authors design the Aesthetic Priors Meta-Learner (APML), a dual-core architecture with a large-scale knowledge backbone (MuseNet-XL) and a lightweight aesthetic core (FEP-RNN). APML is trained via a three-phase meta-learning process, optimizing "aesthetic priors" to enable few-shot music style transfer.

The authors construct the Olympus-Meta Pipeline, a new benchmark with 150 distinct music style tasks, and validate APML against 10 state-of-the-art baselines. Experimental results show APML achieves state-of-the-art performance in stylistic consistency (FMD reduced by 12.5% vs. strongest baseline) and theoretical alignment (FEDR/CAV improved by 17.6%/16.2%), with rigorous ablation studies confirming the necessity of the free energy objective and dual-core design.

**Strengths:**

Translating abstract aesthetic experience into a computable objective (free energy reduction rate) by integrating FEP (computational neuroscience) and meta-learning (machine learning) is a creative cross-disciplinary contribution, avoiding the "statistical mimicry" of prior generative models.

The authors use a comprehensive evaluation suite (11 metrics covering accuracy, style consistency, and theory validation) and compare against 10 diverse baselines, ensuring results are credible. Ablation studies (objective function, model components) isolate the impact of key design choices, and robustness tests (noise injection, cross-domain input) demonstrate model resilience.

The work moves beyond "generating music" to "generating aesthetically pleasing music," addressing a critical limitation of current creative AI. It also provides a computational framework for neuroaesthetics, linking subjective experience to objective model dynamics.

**Weaknesses:**

This paper lack human preference validation. While the paper uses theory-driven metrics (FEDR/CAV) to proxy aesthetic experience, it does not include large-scale human evaluations (e.g., user studies rating "aesthetic appeal" of generated music). This limits the direct link between computational metrics and subjective human perception.

This paper lack the discussion of computational efficiency.  APML’s meta-learning phase (especially the MAML-like inner/outer loops) and FEP-RNN’s GM-based free energy calculation may be computationally expensive. The paper does not analyze inference speed or parameter efficiency, which are important for real-world deployment.

The model and benchmark only focus on piano-centered midi, lead to serious restrictions.

The datasets are mainly western pop, rock, jazz, and folk musics, lack other national musics and new-style musics like dj and hiphop musics.

**Questions:**

The paper states that APML’s initial FEP-RNN weights embody "aesthetic priors," but how can these priors be interpreted? For example, do they encode specific musical rules (e.g., harmonic "tension-release") or more abstract patterns? Could the authors visualize or quantify these priors to strengthen their connection to human aesthetic intuition?

---

### Official Review · Reviewer_KCth · 2025-10-31

**Soundness:** 2
**Presentation:** 1
**Contribution:** 2
**Rating:** 2
**Confidence:** 2

**Summary:**

This paper proposes to quantify the musical pleasance with "free energy reduction"(rate of learning) inspired from neuroscience. Then the authors propose dual-components: Knowledge Backbone (MuseNet-XL based on Music Transformer) and Aesthetic Core (called FEP-RNN based on GRU). To train them, it introduces three-phase training. Especially, the last phase is Aesthetic Priors Meta-Learning, utilize MAML-like meta learning to learn aesthetic priors. To demonstrate the effectiveness of the proposed approach, the authors first construct a MIDI music style transfer benchmark dataset. Then, in experiment, it is shown that the proposed approach outperforms baselines.

**Strengths:**

- This paper proposes an interesting and fresh perspective of quantifying aesthetic pleasure of music with a theory.
- The authors create a new music style transfer benchmark, which might be useful to the community.
- In experiment, it is shown that the proposed approach outperforms baselines.

**Weaknesses:**

- Writing is poor and it is hard to read and understand. The paper has very short introduction and related work, and it is hard to understand the reasoning of the proposed approach from the given goal.
- It is not clear why the proposed approach of meta learning and thus finding good aesthetic priors is related to prove the theory proposed. Do we need meta-learning, and can you just try to learn a model that can generate more aesthetically appealing result without meta learning? In addition, it is also unclear why music style transfer is good to demonstrate the proposed theory.
- The proposed approach was only evaluated on the newly introduce benchmark by the authors, but it is desirable to evaluate on more datasets like the ones used in the baselines. Furthermore, it is not clear why it is needed to create a new benchmark. What is the difference of the dataset compared to prior ones for music style transfer?

**Questions:**

Additional questions:
- Is it possible to apply the same principe to other domains like image or video generation?
- Is there any reason why you showed only some of the evaluation metrics, not all of them, in the ablation study (Table 2)?

Please address the weaknesses and questions raised above.

---

### Official Review · Reviewer_7n6D · 2025-11-01

**Soundness:** 1
**Presentation:** 2
**Contribution:** 2
**Rating:** 0
**Confidence:** 4

**Summary:**

This paper presents an aesthetic prior for generative music modeling, and a meta learner for few-shot stylistic symbolic music generator.

**Strengths:**

The idea of formulating aesthetic prior is useful for music modeling, and the idea of iterative update of the inner (stylistic parameters) and outer (aesthetic prior) loop is interesting.

**Weaknesses:**

1. The paper lacks computational musicology evidence of the aesthetic prior. The only reference is from cognitive science, but more evidence is needed for computational modeling.
2. The model fails to cite or compare with other novelty-based generative methods, including [1-2]. Theoretical insights on the difference between these methods and this paper are absent.
3. No subjective evaluation on the musicality.
4. The case studies in Fig. 3 and appendix are very problematic. First of all, all samples seem to be simple melody + chords, which are far from enough to model 100+ distinct styles. Also in Fig. 3 (middle), the chord of the APML output seems to be static, which makes the "resolution" peaks in Fig. 3 (bottom) very strange. The generated melody also seems very random and does not show a meaningful cadence or other ways of resolution. All these demos make me concern about the model's musicality issue, and no subjective evaluation makes it more problematic.

[1] Chen, Y. W., Lee, H. S., Chen, Y. H., & Wang, H. M. (2021). SurpriseNet: Melody harmonization conditioning on user-controlled surprise contours. arXiv preprint arXiv:2108.00378.

[2] Thickstun, J., Hall, D., Donahue, C., & Liang, P. (2023). Anticipatory music transformer. arXiv preprint arXiv:2306.08620.

**Questions:**

1. In Eqn. (7), why is it $q (c_t|c_{\leq t})$?
2. In 5.2, what are the 150 musical styles? A list is needed. How do you guarantee that there is no major overlapping between the training and testing split?

---

### Meta-Review · Area_Chair_kSUT · 2026-01-07

**Summary:**

I balieve that, in general, applying the free energy principle/active inference to define a meaningful prior over music generation would be an interesting direction. However, I do not think this paper accomplishes that goal, and largely concur with the critiques levied by the reviewers:
- **7n6D**: “lacks computational musicology evidence of the aesthetic prior”; “fails to cite or compare with other novelty-based generative methods”; “[lack of] subjective evaluation”; qualitative issues with examples
- **KCth**: writing quality; necessity/relatedness of meta-learning approach; limited evaluation
- **bhj6**: lack of human validation; limited focus on piano-centered MIDI output
- **f2bm**: “narrow definition of aesthetics”; lack of human validation

In addition to this, I find several claims in the paper to be unsupported, e.g.:
1. No application of FEP in AI: this ignores the very deep connection between FEP and predictive coding (which this paper itself makes use of), statistical surprisal, etc. Lots of papers depend on this (e.g. contrastive predictive coding methods, etc.)
2. Many of the motivation claims are so underpsecified as to be difficult to even pin down, let alone disagree with. The opening sentence of the introduction does not explain what precisely is meant by “aesthetic compass” but states as fact that modern generative models lack them (how would we know, if there isn’t a consensus on what this object is?). That models “mimic the statistics of a style without grasping the underlying principles” for which no evidence is provided.

The paper also fails to cite or make meaningful use of much important and influential work on applications of the FEP to the study of music and musical aesthetics, including several papers involving K. Friston himself.

**Reviewer Concerns:**

There appears to have been no rebuttal submitted.

**Reviewer Scores:**

Since no rebuttal was submitted, the reviewers haven’t been given reasonable cause to adjust their scores.

---

### Decision · Program_Chairs · 2026-01-26

Reject